# Responses of dissolved organic carbon to freeze–thaw cycles associated with the changes in microbial activity and soil structure

**You Jin Kim, Jinhyun Kim, and Ji Young Jung**

Division of Life Sciences, Korea Polar Research Institute, 26 Songdomirae-ro, Yeonsu-gu, Incheon 21990, Republic of Korea

**Correspondence:** Ji Young Jung (jyjung@kopri.re.kr)

**Abstract.** Arctic warming accelerates snowmelt, exposing soil surfaces with shallow or no snow cover to freeze–thaw cycles (FTCs) more frequently in early spring and late autumn. FTCs influence Arctic soil C dynamics by increasing or decreasing the amount of dissolved organic carbon (DOC); however, mechanism-based explanations of DOC changes that consider other soil biogeochemical properties are limited. To understand the effects of FTCs on Arctic soil responses, we designed microcosms with surface organic soils from Alaska and investigated several soil biogeochemical changes for seven successive temperature fluctuations of freezing at $-9.0 \pm 0.3\,°C$ and thawing at $6.2 \pm 0.3\,°C$ for 12 h each. FTCs significantly changed the following soil variables: soil $CO_2$ production ($CO_2$), DOC and total dissolved nitrogen (TDN) contents, two DOC quality indices ($SUVA_{254}$ and $A_{365} / A_{254}$), microaggregate (53–250 μm) distribution, and small-sized mesopore (0.2–10 μm) proportion. Multivariate statistical analyses indicated that the FTCs improved soil structure at the scale of microaggregates and small-sized mesopores, facilitating DOC decomposition by soil microbes and changes in DOC quantity and quality by FTCs. This study showed that FTCs increased soil $CO_2$ production, indicating that FTCs affected DOC characteristics without negatively impacting microbial activity. Soil microaggregation enhanced by FTCs and the subsequent increase in microbial activity and small-sized pore proportion could promote DOC decomposition, decreasing the DOC quantity. This study provides a mechanism-based interpretation of how FTCs alter DOC characteristics of the organic soil in the active layer by incorporating structural changes and microbial responses, improving our understanding of Arctic soil C dynamics.

## 1 Introduction

Arctic tundra soils store approximately $1300 \pm 200$ Pg of soil organic carbon (SOC) in permafrost (Čapek et al., 2015), which accounts for approximately 30 % of the global SOC pool (Xu et al., 2009). Recently, Arctic warming, which occurs 4 times faster than global warming (Rantanen et al., 2022), has enhanced permafrost thaw, causing the previously stored SOC to be released into greenhouse gases ($CO_2$ and $CH_4$) and/or leaching dissolved organic carbon (DOC) (Estop-Aragonés et al., 2020). In particular, DOC released from the active layer could be further decomposed by soil microorganisms into $CO_2$ and $CH_4$, leading to a positive feedback on permafrost thawing (Foster et al., 2016; Yi et al., 2015). Permafrost thaw also influences the Arctic watershed through the export of terrestrial-derived DOC into the surrounding lakes and seas (Al-Houri et al., 2009). The exported DOC from the active layer can horizontally migrate along the unfrozen vicinity between frozen layers; in addition, it can infiltrate the deeper active layer and upper permafrost during the thawing phase (Ban et al., 2016; Han et al., 2018). Thus, the measurement of quantitative and qualitative DOC changes are necessary for understanding the response of permafrost C dynamics to Arctic warming (Xu et al., 2009; Foster et al., 2016; Perez-Mon et al., 2020).

Moreover, increased temperatures in Arctic regions accelerate snowmelt (Henry, 2008; Førland et al., 2011; Kreyling et al., 2008) and cause rainfall instead of snowfall (Henry, 2013; IPCC, 2014), leading to the absence of snow cover on the soil surface (Callaghan et al., 1998; Heal et al., 1998). In Arctic regions, snow plays a key role in protecting tundra soils against dramatic temperature changes caused by harsh climates (Royer et al., 2021). Exposed soil surfaces lacking snow cover are likely to undergo more frequent freeze–thaw

cycles (FTCs) in the early spring and late autumn because they are directly influenced by the diurnal fluctuations of atmospheric temperature (Kreyling et al., 2008; Henry, 2013; Freppaz et al., 2007). Climate models project that the air temperature of the Arctic may continue to rise owing to climate change, thereby enhancing the occurrence of FTCs in permafrost soils within the near future (Henry, 2008; Groffman et al., 2011; Grogan et al., 2004).

Numerous studies have reported influences of FTCs on the labile soil C content, which is strongly related to microbial activity in Arctic tundra soils (Sawicka et al., 2010; Schimel and Clein, 1996; Larsen et al., 2002; Männistö et al., 2009; Lipson and Monson, 1998; Perez-Mon et al., 2020; Grogan et al., 2004; Foster et al., 2016; Schimel and Mikan, 2005; Sjursen et al., 2005; Yi et al., 2015). FTCs have been reported to increase the amount of DOC in a few tundra and non-tundra soils, attributed to a decrease in microbial utilization of DOC due to cell lysis generally occurring below $-7$ to $-11\,°C$ of freezing temperature (D. Gao et al., 2018, 2021; Song et al., 2017; Schimel and Clein, 1996; Larsen et al., 2002). In contrast, several studies have shown negligible changes or decreases in the amount of DOC by FTCs, without negative responses of microbial biomass, community, and enzymatic activities (Männistö et al., 2009; Lipson and Monson, 1998; Perez-Mon et al., 2020). This was interpreted as soil microorganisms in the Arctic tundra having already adapted to extreme temperature fluctuations for a long period of time; thus, FTCs could not inhibit microbial DOC utilization for growth and activity compared to non-tundra regions, such as forest, grassland, and cropland (D. Gao et al., 2018, 2021; Song et al., 2017). These controversial results suggest that further research and evidence of DOC changes by FTCs are required for an improved mechanism-based understanding of tundra soil C dynamics in the early spring and late autumn.

FTCs can indirectly affect the Arctic tundra DOC dynamics through soil structural changes such as the fragmentation, rearrangement, and aggregation of soil particles (Matzner and Borken, 2008; Zhang et al., 2016). Owing to the phase transitions in soil water during FTCs, soil matrix cracks and the physical degradation of soil aggregates have been reported in previous studies (Oztas and Fayetorbay, 2003; Wang et al., 2012; Hall and André, 2003). In contrast, several researchers have found that FTCs enhance soil aggregate stability (Lehrsch, 1998) and small-sized aggregate formation (50–250 and 500–1000 μm) (Li and Fan, 2014). Changes in soil aggregate distribution by FTCs likely affect the soil pore volume and spatial distribution (Lu et al., 2021; Al-Houri et al., 2009; Oztas and Fayetorbay, 2003; Viklander, 1998), leading to alterations in soil water retention and DOC release (Matzner and Borken, 2008; Song et al., 2017; D. Gao et al., 2018; Feng et al., 2007). Additionally, these soil structural changes may affect microbe-mediated soil C mineralization and utilization by improving the soil water and nutrient distribution (Athmann et al., 2013; Liang et al., 2019; Sander

and Gerke, 2007). However, the linkage of how structural changes caused by FTCs, such as the formation of aggregates and pores with specific sizes, affect DOC changes has not been well understood.

This study aimed to identify the effects of FTCs on Arctic tundra DOC dynamics using surface organic soils from the Alaskan tundra undergoing temperature fluctuation during the early spring. We designed two parallel microcosms simulating the FTCs of the study site for two different purposes. One set of microcosms was established for destructive sampling to investigate the temporal changes in soil $CO_2$ production, soil enzyme activities, DOC characteristics, and soil aggregate size distribution. The other set of soil core incubation was prepared with repacked soil to measure pore size distribution (PSD) using soil water retention curves. We tested the following hypotheses: (1) FTCs alter DOC quantity and quality without decreasing the activities of soil microbes previously adapted to temperature fluctuations in the Arctic, and (2) soil aggregate distribution influenced by FTCs changes DOC characteristics by enhancing microbial activities and altering specific-sized soil pore proportion.

## 2 Materials and methods

### 2.1 Site description and soil preparation

Soil samples for microcosm incubation were collected from the moist acidic tundra in Council (64.51° N, 163.39° W) on the Seward Peninsula in Northwest Alaska. The average temperature and precipitation over the past 30 years (1981–2020) are $-3.1\,°C$ and 258 mm (Alaska Climate Research Center). In the early spring (April to May), the minimum and maximum temperatures are $-8.5$ and $7.1\,°C$, respectively (Alaska Climate Research Center). This site is in the discontinuous permafrost zone and tussock tundra dominated by cotton grasses (*Eriophrum vaginatum*), blueberries (*Vaccinium uliginosum*), lichen, and moss (*Sphagnum* spp.) (Kim et al., 2016).

Soil sampling was performed at three random points under similar vegetation compositions. Each point was within a distance of approximately 100 m from each other. At the time of sampling (early July 2010), the active layer depth was approximately 50 cm TS1, measured by a steel rod (1 m). Soil samples were acquired by hammering a stainless steel pipe (7.6 cm diameter × 50 cm long) into the partially or well-degraded organic layer (Oe), mixed with soil minerals, after removing the litter layer (Oi) on the surface. The soil samples were stored at $-20\,°C$ before initiating microcosm incubation. The frozen soil was thawed at $< 4\,°C$, and the surface organic soils were passed through a 2 mm sieve and homogenized by hand. Figure 1 shows the soil sampling and preparation procedure. Soil textural analysis was conducted by a wet sieving and pipette method (Kim et al., 2022). Soil bulk density (BD) was determined by calculating the soil dry weight

contained in the soil core volume. Volumetric water content (VWC) in the soil was measured using a portable sensor with an accuracy of $\pm 0.01\,\mathrm{cm^3\,cm^{-3}}$ (ProCheck Decagon Devices, Washington, USA). Total carbon (C) and nitrogen (N) contents were determined through combustion (950 °C) with an elemental analyzer (vario MAX cube; Elementar vario MAX cube; Elementar, Langenselbold, Germany). The basic soil properties are summarized in Table S1 in the Supplement.

## 2.2   Soil incubation with freeze–thaw cycles

Soil incubation was conducted with two parallel sets of microcosms: one for destructive sampling and the other for monitoring soil PSD changes (Fig. 1). The destructive sampling set was established using a 380 mL polypropylene bottle to investigate the soil biogeochemical properties influenced by FTCs. The other microcosm set was created by reconstructing the small-sized soil core (5 cm diameter × 5 cm long) to compare PSD alterations under incubation conditions with/without the impact of FTCs. We established FTC and CON as experimental groups. The FTC is a treatment with seven successive temperature fluctuations of freezing at $-9.0 \pm 0.3$ °C and thawing at $6.2 \pm 0.3$ °C for 12 h each (Fig. S1 in the Supplement). The meta-analysis and several other studies showed that soil carbon dynamics (Gao et al., 2021) and total porosity (Liu et al., 2021; Ma et al., 2021) responded to 3–10 FTCs; thus, seven successive FTCs were adopted in this study. CON is a control group that maintained an average temperature of $-2$ °C without any fluctuations (Fig. S1). The freezing and thawing temperatures in the FTC treatment corresponded to the early spring conditions observed at the study site. We ensured that our FTC treatment was adequate for complete freezing and thawing of the soil based on previous studies conducted under similar conditions (Freppaz et al., 2007; Larsen et al., 2002; Song et al., 2017; Han et al., 2018). Three replicates were used for the CON and FTC treatments of each microcosm set. In all incubation sets, the initial soil BD was adjusted to $0.72\,\mathrm{g\,cm^{-3}}$, similar to field-soil conditions (Fig. 1; Table S1). For the destructive sampling set, we used 260 g of homogenized fresh soil (154.7 g dry weight) to a soil volume of $215\,\mathrm{cm^3}$. The microcosm set using the small-sized cores was established with 120 g of homogenized fresh soil (71.5 g dry weight) in a $99\,\mathrm{cm^3}$ volume. The VWC for all incubation soils was also standardized by spraying water using a pipette to $0.50\,\mathrm{cm^3\,cm^{-3}}$ (70 % water-filled pore space), a similar level to field soils (Fig. 1; Table S1).

## 2.3   Soil analyses

All soil analyses, except for the PSD measurement, were conducted using the first incubation set for destructive sampling. Soil $CO_2$ production ($\overline{CO_2}$), widely accepted as a proxy for overall microbial activity (Kim and Yoo, 2021; Maikhuri and Rao, 2012; Davidson et al., 1998; Kuzyakov and Domanski, 2000; Lipson and Schmidt, 2004; Raich and Schlesinger, 1992), was estimated by daily measurement of the $CO_2$ flux from soil incubation during the entire incubation period. We collected gas samples from the headspace through a septum using 10 mL syringes (BD Luer-Lok tip, BD Company, Franklin Lakes, NJ, USA) before and after sealing the incubation bottle for 60 min. The gas samples were analyzed using gas chromatography (Agilent 7890A, Santa Clara, CA, USA) with a hydrogen flame ionization detector to determine the $CO_2$ concentration. The $CO_2$ flux was calculated based on changes in headspace concentration over 60 min using the following Eq. (1) (Troy et al., 2013):

$$CO_2 \text{ flux} = \frac{\mathrm{dGas}}{\mathrm{d}t} \times \frac{V}{A} \times \frac{[P \times 100 \times \mathrm{MW}]}{R} \times \frac{273}{T}, \quad (1)$$

where $\mathrm{dGas}/\mathrm{d}t$ is the change in the $CO_2$ concentration before and after sealing the incubation bottle for 60 min, $V$ and $A$ are the volume and area of the incubation bottle, $P$ is the atmospheric pressure (1 atm), MW is the molecular weight of $CO_2$ ($44.01\,\mathrm{g\,mol^{-1}}$), $R$ is a gas constant ($0.082\,\mathrm{atm\,L\,mol^{-1}\,K^{-1}}$), and $T$ is the absolute temperature during gas collection (293 K). In addition, we calculated the mean $CO_2$ production ($\overline{CO_2}$) by averaging the daily measured $CO_2$ flux during the entire incubation period.

Several soil biogeochemical properties were measured at the end of seven successive FTCs. Soil extracellular enzyme activity was determined: two oxidases (peroxidase and phenol-oxidase) and four hydrolases ($\beta$-glucosidase, cellobiase, N-acetyl-$\beta$-glucosaminidase, and aminopeptidase) involved in soil C and N cycling (Liao et al., 2022) were identified. These enzyme activities were measured by fluorometric assays using L-3,4-dihydroxyphenylalanine (L-DOPA) solution for oxidases and methylumbelliferyl (MUF)-linked substrates for hydrolases (Kwon et al., 2013).

To quantify the available C and N in soils, we measured the DOC and total dissolved nitrogen (TDN) content via water extraction. After adding 40 mL of distilled water, 20 g of fresh soil was shaken for 1 h, centrifuged, and filtered with a 0.45 μm filter to obtain supernatant. The supernatants were measured using a Multi N/C 3100 analyzer (Analytik Jena, Jena, Thüringen, Germany). The filtered samples were also used to estimate DOC qualitative indices. The specific ultraviolet absorbance at 254 nm ($SUVA_{254}$), which allowed for the estimation of DOC aromaticity, was calculated using UV absorbance at 254 nm ($A_{254}$) divided by DOC concentration ($\mathrm{mg\,C\,L^{-1}}$) and the path length (m) of the UV cuvette of the spectrometer (Eppendorf, Hamburg, Germany) (Lim et al., 2021). The ratio of $A_{254}$ to $A_{365}$ ($A_{254} / A_{365}$) was used as a proxy that is negatively related to the molecular weight of the DOC compounds (Berggren et al., 2018). For $NH_4^+$-N and $NO_3^-$-N content analysis, 5 g of fresh soil was shaken with a 2 M KCl solution for 1 h, centrifuged, and filtered through Whatman #42 paper. The filtrates were analyzed using an

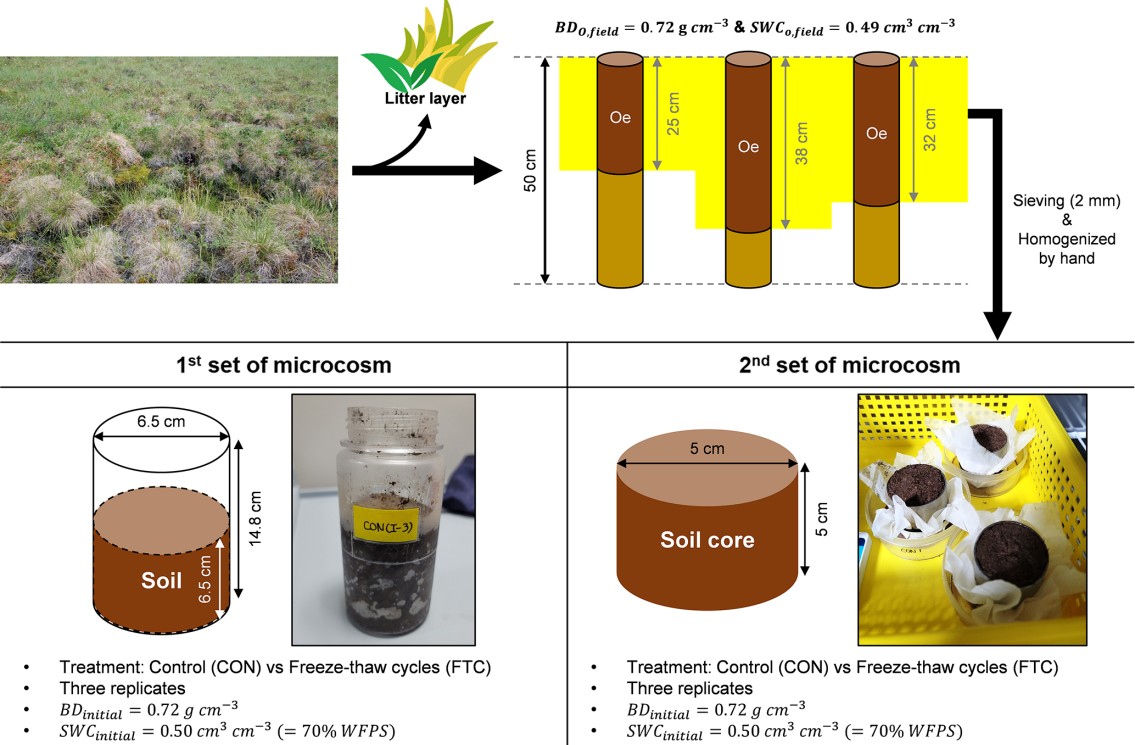

**Figure 1.** Soil sampling and experimental design for the microcosm incubation study.

auto-analyzer (QuAAtro, SEAL Analytical GmbH, Norderstedt, Schleswig-Holstein, Germany).

Soil aggregate fractionation was performed using density separation and a subsequent wet sieving method at the end of incubation (Kim et al., 2021; Yoo et al., 2017). Then, 20 g of air-dried and 2 mm sieved soil was mixed with 35 mL of distilled water for 30 min. The soil–water mixture was left overnight and then centrifuged at 3200 rpm for 10 min. The supernatant, which was a free-light fraction ($< 1.0\,\mathrm{g\,cm^{-3}}$), was collected using precombusted glass microfiber filters (GF/A). The heavy fraction was wet-sieved using 1000, 250, and 53 µm sieves to separate water-stable aggregates into four size classes: mega-aggregates (1000–2000 µm), macroaggregates (250–1000 µm), microaggregates (53–250 µm), and mineral-associated fractions ($< 53$ µm). The wet sieving procedure was performed by manually shaking each sieve 100 times for more than 2 min. All aggregate fractions remaining on the GF/A filters and sieves were transferred to an aluminum dish, dried in an oven at 60 °C for a week, and then weighed.

To estimate the PSD on the post-incubation core soils (5 cm diameter × 5 cm long), soil water release curves were generated by the Hydrus-1D model equipped with van Genuchten soil-hydraulic equations, which can be applied to organic and mineral soil (Šimůnek et al., 2013; Dettmann et al., 2014). The modeling procedure requires the van Genuchten parameters, calculated using volumetric water content at field capacity and wilting point (Kameyama et al., 2012; Likos et al., 2014). The volumetric water content at field capacity was measured by the soil water content at a matric potential of $-33$ kPa using a sand box (Eijkelkamp Agrisearch Equipment, Santa Barbara, CA, USA) (Yoo et al., 2020), after saturating soil core samples. As the matric potential used in a sandbox did not sufficiently cover the entire soil water release curves, the volumetric water content at the wilting point ($-1500$ kPa) was calculated using the pedotransfer function from soil carbon content and bulk density (da Silva and Kay, 1997). Lastly, the PSD was estimated from the matric potential corresponding to each pore size using the Young–Laplace equation (Kim et al., 2021).

## 2.4 Statistical analyses

Analysis of variance (ANOVA) was performed using the generalized linear model (GLM) procedure (SAS 9.4, SAS Institute Inc., Cary, NC, USA) to compare the measurement data between the CON and FTC treatments. Least square means were used to assess significant differences among treatments at $p < 0.05$. Following ANOVA, we performed a principal component analysis (PCA) using the "FactoMineR" package in RStudio 4.2.1 (Rstudio Inc., Boston, MA, USA) to verify whether soil variables with significant responses could discriminate the FTC soil from the CON soil. The Pearson's correlation analysis was conducted using the CORR procedure (SAS 9.4) to examine the relationship between soil

**Table 1.** Characteristics of dissolved organic matter (DOC, TDN, SUVA$_{254}$, A$_{254}$ / A$_{365}$) and inorganic nitrogen (NO$_3^-$-N and NH$_4^+$-N) contents in the soils treated (FTC) and non-treated (CON) by freeze–thaw cycles.

| | DOC (mg kg$^{-1}$ soil) | TDN | SUVA$_{254}$ (L mg$^{-1}$ m$^{-1}$) | A$_{254}$ / A$_{365}$ | NO$_3^-$-N (mg kg$^{-1}$ soil) | NH$_4^+$-N |
|---|---|---|---|---|---|---|
| CON | 659.91 (3.10) | 39.01 (0.82) | 1.81 (0.06) | 4.02 (0.07) | 0.03 ($<$ 0.01) | 0.01 ($<$ 0.01) |
| FTC | 467.04 (2.93) | 25.44 (0.30) | 2.93 (0.14) | 3.72 (0.01) | 0.03 ($<$ 0.01) | 0.02 (0.01) |
| $p$ value | $<$ 0.001** | $<$ 0.001** | 0.002** | 0.016** | 0.328 | 0.340 |
| $F$ value | 2046.21 | 241.87 | 56.21 | 16.38 | 1.24 | 1.17 |

Note: The asterisks ** and * indicate significant differences between treatments at the $p < 0.05$ and $p < 0.10$ levels, respectively. The numbers in parentheses are standard errors ($n = 3$).

**Table 2.** Mean CO$_2$ production ($\overline{CO}_2$) and extracellular enzyme activities in the FTC and CON soils

| | $\overline{CO}_2$ (mg m$^{-2}$ h$^{-1}$) | Peroxidase (μmol g$^{-1}$ soil min$^{-1}$) | Phenol-oxidase | $\beta$-glucosidase | Cellobiase (nmol g$^{-1}$ soil min$^{-1}$) | N-acetyl-$\beta$-glucosaminidase | Aminopeptidase |
|---|---|---|---|---|---|---|---|
| CON | 3.65 (2.11) | 15.09 (0.81) | 1.26 (0.49) | 0.082 (0.008) | 0.137 (0.004) | 0.076 (0.003) | 0.786 (0.034) |
| FTC | 42.54 (6.33) | 15.00 (1.68) | 0.99 (0.05) | 0.073 (0.002) | 0.139 (0.003) | 0.074 (0.001) | 0.781 (0.016) |
| $p$ value | 0.004** | 0.964 | 0.607 | 0.607 | 0.765 | 0.623 | 0.909 |
| $F$ value | 33.96 | $<$ 0.01 | 0.31 | 1.06 | 0.10 | 0.28 | 0.01 |

Note: The asterisks ** and * indicate significant differences between treatments at the $p < 0.05$ and $p < 0.10$ levels, respectively. The numbers in parentheses are standard errors ($n = 3$).

variables. Finally, multiple linear regression (MLR) analyses were employed to illustrate the mechanisms by which the FTC-influenced soil biogeochemical variables contributed to changes in DOC characteristics. Based on the results of ANOVA, PCA, and the Pearson's correlation analysis, we extracted all plausible interactions among soil physical and biogeochemical variables significantly affected by FTCs and refined these interactions by finding the best-fitting regression sets from MLR. The MLR analyses were generated using SigmaPlot 13.0.

## 3 Results

### 3.1 Soil biogeochemical and structural changes by freeze–thaw cycles

The quantity and quality of DOC in the soil solution were altered by FTCs, as presented in Table 1. The FTC soil exhibited lower DOC and TDN contents by 29 % and 35 %, respectively, compared to the CON soil ($p < 0.001$). As proxies for DOC quality, SUVA$_{254}$ was higher ($p = 0.002$), but A$_{254}$ / A$_{365}$ was lower ($p = 0.016$) in the FTC soil than in the CON soil. The increase in SUVA$_{254}$ indicates an increase in the aromaticity of DOC (Lim et al., 2021), while the decrease in A$_{254}$ / A$_{365}$ reflects an increase in the molecular weight of DOC (Berggren et al., 2018). In contrast, no significant changes in inorganic N (NH$_4^+$-N and NO$_3^-$-N) content were determined with FTCs ($p > 0.100$).

The mean CO$_2$ production ($\overline{CO}_2$) in the FTC soil was 42.54 mg m$^{-2}$ h$^{-1}$, 12 times higher than that in the CON soil

(3.65 mg m$^{-2}$ h$^{-1}$, $p = 0.004$), as shown in Table 2. This is because the $\overline{CO}_2$ in FTC soil was significantly higher than in CON soil from the early stages of FTCs ($p < 0.05$) and remained consistently higher until the end of the incubation (Fig. S2). Conversely, no significant differences ($p > 0.10$) were observed between treatments in all types of microbial extracellular enzyme activities (Table 2).

The FTC resulted in minor differences in the mass proportion of microaggregates (53–250 μm) and mineral-associated fractions ($<$ 53 μm), which account for an average of 35 % and 34 % of the total in each soil (Table 3). The mass proportion of microaggregates marginally increased by 17 % in the FTC soil compared to that in the CON soil ($p = 0.066$). Although the mineral-associated fractions were insignificantly reduced by FTCs ($p = 0.257$), the reduction level (18 %) corresponded to the increased distribution of microaggregates by FTCs. Moreover, FTCs caused a significant difference in the PSD, particularly in the small-sized mesopores (0.2–10 μm), which accounted for 44 %–45 % TS3 of the total soil pores (Table 4), as estimated using water retention curves (Fig. S3). Despite the small magnitude of difference, the proportion of small-sized mesopores in the FTC soil exhibited a statistically significant increase compared to that in the CON soil ($p = 0.024$).

### 3.2 Influencing variables deriving dissolved organic carbon changes by freeze–thaw cycles

The PCA was used to further identify relationships among seven soil variables that showed significant responses to

**Table 3.** Aggregate size–density distribution in the FTC and CON soils.

| | Free-light fraction | Heavy fractions (water-stable aggregates) | | | |
| --- | --- | --- | --- | --- | --- |
| | | Mega-aggregate (1000–2000 µm) | Macroaggregate (250–1000 µm) | Microaggregate (53–250 µm) | Mineral-associated fraction (< 53 µm) |
| | | ($g\,100\,g^{-1}$ soil) TS2 | | | |
| CON | 0.58 (0.04) | 7.01 (0.83) | 23.54 (0.67) | 32.07 (1.11) | 37.03 (3.84) |
| FTC | 0.76 (0.08) | 6.86 (0.80) | 25.11 (1.60) | 37.45 (1.83) | 30.40 (3.25) |
| $p$ value | 0.114 | 0.900 | 0.416 | 0.066* | 0.257 |
| $F$ value | 4.07 | 0.02 | 0.82 | 6.34 | 1.74 |

Note: The asterisks ** and * indicate significant differences between treatments at the $p < 0.05$ and $p < 0.10$ levels, respectively. The numbers in parentheses are standard errors ($n = 3$).

**Table 4.** Pore size distribution (PSD) in the FTC and CON soils.

| | Total soil pore volume | Pore volume among the different-sized classes | | | |
| --- | --- | --- | --- | --- | --- |
| | | Macropore (> 30 µm) | Mesopore | | Micropore (< 0.2 µm) |
| | | | Large (10–30 µm) | Small (0.2–10 µm) | |
| | | ($cm^3\,g^{-1}$ soil) | | | |
| CON | 1.022 (0.021) | 0.287 (0.007) | 0.152 (0.004) | 0.451 (0.006) | 0.133 (0.005) |
| FTC | 1.071 (0.022) | 0.289 (0.007) | 0.156 (0.005) | 0.479 (0.005) | 0.146 (0.005) |
| $p$ value | 0.178 | 0.830 | 0.497 | 0.024** | 0.149 |
| $F$ value | 2.67 | 0.05 | 0.56 | 12.47 | 3.19 |

Note: The asterisks ** and * indicate significant differences between treatments at the $p < 0.05$ and $p < 0.10$ levels, respectively. The numbers in parentheses are standard errors ($n = 3$).

FTCs: DOC and TDN contents, $SUVA_{254}$, $A_{254}/A_{365}$, $\overline{CO}_2$, microaggregates, and small-sized mesopores (Tables 1, 2, 3, and 4). The first two principal components (PCs) accounted for 93.9 % of the total variance, with PC1 clearly clustering the FTC and CON treatments (Fig. 2). PC1 exhibited positive correlations with $SUVA_{254}$, $\overline{CO}_2$, microaggregates, and small-sized mesopores, while it showed negative correlations with DOC, TDN, and $A_{254}/A_{365}$. In Fig. 2, the microaggregate was nearly perpendicular to the DOC and TDN contents, $SUVA_{254}$, and $A_{254}/A_{365}$, indicating a weak or no correlation between them. This is consistent with the results of the Pearson's correlation analysis (Fig. 3). The DOC and TDN contents showed strong correlations with $\overline{CO}_2$ and small-sized mesopores ($p < 0.05$) but had a weaker correlation with microaggregates at a significance level of $p < 0.10$. Furthermore, there were no significant correlations between microaggregates and proxies for DOC quality, including $SUVA_{254}$ and $A_{254}/A_{365}$ ($p > 0.10$). Lastly, a conceptual diagram (Fig. 4) was created using MLR analyses (Table 5) to depict the relationships between soil structural properties, microbial activity, and DOC quantity and quality as influenced by FTCs. In Table 5, $\overline{CO}_2$ and small-sized mesopores were the best-fitting variables for explaining the contents of DOC (adjusted $R^2 = 0.911$,

$p = 0.012$) and TDN (adjusted $R^2 = 0.869$, $p = 0.022$). The variance inflation factors (VIFs) resulting from the MLR analyses were < 10, indicating no collinearity between $\overline{CO}_2$ and small-sized mesopores as independent variables. The variables for best representing $SUVA_{254}$ and $A_{254}/A_{365}$ were $\overline{CO}_2$ (adjusted $R^2 = 0.703$, $p = 0.023$) and small-sized mesopores (adjusted $R^2 = 0.878$, $p = 0.004$), respectively. The addition of microaggregates reduced the adjusted $R^2$ of the best-fitting regression for explaining DOC, TDN, SUVA, and $A_{254}/A_{365}$. Microaggregates correlated with $\overline{CO}_2$ (adjusted $R^2 = 0.557$, $p = 0.054$) and small-sized mesopores (adjusted $R^2 = 0.618$, $p = 0.039$). As a result, we speculated that microaggregates indirectly, rather than directly, affected the quantitative and qualitative DOC variables through its correlation with $\overline{CO}_2$ and small-sized mesopores, as illustrated in Fig. 4.

## 4  Discussion

### 4.1  Effects of freeze–thaw cycles on dissolved organic carbon associated with microbial activities

The seven successive FTCs reduced soil DOC and TDN contents compared to the non-treated condition, aligning with

**Table 5.** Multiple linear regression (MLR) analyses of the FTC and CON soils. The analyses were performed on the observed variables that showed significant differences between treatments. The independent variables were standardized to avoid bias due to the different scales among them.

| Dependent variable | # | $R^2$ | $R^2_{\text{adjust}}$ | $p$ value | Independent variable | | | |
|---|---|---|---|---|---|---|---|---|
| | | | | | Predictor | Standardized $\beta$ coefficient | $p$ value | VIF |
| DOC | 1 | 0.885 | 0.856 | 0.005** | Constant | $-0.664 \times 10^{-15}$ | – | – |
| | | | | | $\text{RES}_{\text{mean}}$ | $-0.941$ | 0.005** | – |
| | 2 | 0.946 | 0.911 | 0.012** | Constant | $-1.116 \times 10^{-15}$ | – | – |
| | | | | | $\text{RES}_{\text{mean}}$ | $-0.639$ | 0.056* | 2.5 |
| | | | | | Small-sized mesopore | $-0.390$ | 0.162 | 2.5 |
| | 3 | 0.951 | 0.878 | 0.072* | Constant | $-1.513 \times 10^{-15}$ | – | – |
| | | | | | $\text{RES}_{\text{mean}}$ | $-0.695$ | 0.129 | 3.1 |
| | | | | | Small-sized mesopore | $-0.464$ | 0.260 | 3.4 |
| | | | | | Microaggregate | 0.140 | 0.702 | 4.1 |
| TDN | 1 | 0.896 | 0.870 | 0.004** | Constant | $0.017 \times 10^{-15}$ | – | – |
| | | | | | $\text{RES}_{\text{mean}}$ | $-0.947$ | 0.004** | – |
| | 2 | 0.922 | 0.869 | 0.022** | Constant | $-0.274$ | – | – |
| | | | | | $\text{RES}_{\text{mean}}$ | $-0.752$ | 0.060* | 2.5 |
| | | | | | Small-sized mesopore | $-0.251$ | 0.397 | 2.5 |
| | 3 | 0.928 | 0.821 | 0.011** | Constant | $-0.746 \times 10^{-15}$ | – | – |
| | | | | | $\text{RES}_{\text{mean}}$ | $-0.818$ | 0.135 | 3.1 |
| | | | | | Small-sized mesopore | $-0.339$ | 0.446 | 3.6 |
| | | | | | Microaggregate | 0.167 | 0.707 | 4.1 |
| $\text{SUVA}_{254}$ | 1 | 0.763 | 0.703 | 0.023** | Constant | $1.359 \times 10^{-15}$ | – | – |
| | | | | | $\text{RES}_{\text{mean}}$ | 0.873 | 0.023** | – |
| | 2 | 0.809 | 0.681 | 0.084* | Constant | $1.751 \times 10^{-15}$ | – | – |
| | | | | | $\text{RES}_{\text{mean}}$ | 0.612 | 0.222 | 2.5 |
| | | | | | Small-sized mesopore | 0.338 | 0.458 | 2.5 |
| $\text{A}_{254} / \text{A}_{365}$ | 1 | 0.902 | 0.878 | 0.004** | Constant | $-2.072 \times 10^{-15}$ | – | – |
| | | | | | Small-sized mesopore | $-0.950$ | 0.004** | – |
| | 2 | 0.920 | 0.866 | 0.023* | Constant | $-1.941 \times 10^{-15}$ | – | – |
| | | | | | Small-sized mesopore | $-0.790$ | 0.055* | 2.5 |
| | | | | | $\text{RES}_{\text{mean}}$ | $-0.207$ | 0.481 | 2.5 |
| $\overline{\text{CO}_2}$ | 1 | 0.646 | 0.557 | 0.054* | Constant | $-2.019$ | – | – |
| | | | | | Microaggregate | 0.804 | 0.054 | – |
| | 2 | 0.681 | 0.469 | 0.180 | Constant | $-0.975$ | – | – |
| | | | | | Microaggregate | 0.521 | 0.442 | 3.3 |
| | | | | | Small-sized mesopore | 0.340 | 0.605 | 3.3 |
| Small-sized mesopore | 1 | 0.694 | 0.618 | 0.039** | Constant | $-3.077$ | – | – |
| | | | | | Microaggregate | 0.833 | 0.039 | – |

Note: $R^2$ denotes coefficient of determination; $R^2_{\text{adjust}}$ denotes adjusted $R^2$; VIF denotes variance inflation factor; $\overline{\text{CO}_2}$ is the mean $CO_2$ production. The asterisks ** and * indicate significant differences between treatments at the $p < 0.05$ and $p < 0.10$ levels, respectively.

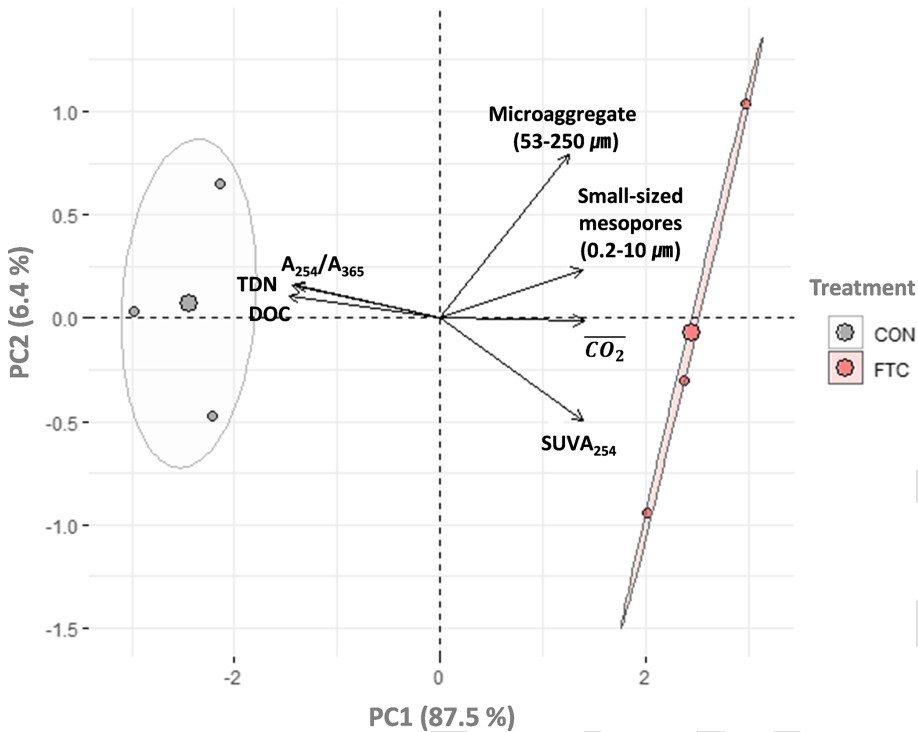

**Figure 2.** Principal component analysis (PCA) for the FTC and CON soils ($n = 3$). The analysis was performed on the observed variables that showed significant differences between treatments. The input variables were standardized to avoid bias due to the different scales among them. Each arrow to the direction of increase for a given variable and its length indicate the strength of the correlation between the variable and ordination scores. Ellipses show confidence intervals of 95 % for each treatment.

the expectation that the quantitative characteristics of DOC were significantly affected by FTCs (Table 1). These results indicate that FTCs can accelerate the microbial decomposition of labile organic matter (Grogan et al., 2004; Han et al., 2018; Foster et al., 2016; Gao et al., 2021). A proxy for overall microbial activity, $CO_2$ production, remained high throughout the incubation under the influence of FTCs (Fig. S2; Table 2). The main reason for our findings likely results from the soil microbial characteristics in the Arctic tundra. In other words, soil microorganisms have already adapted to the frequent temperature fluctuations in early spring and late autumn in the Arctic tundra (Perez-Mon et al., 2020; Koponen and Bååth, 2016; Walker et al., 2006; Song et al., 2017). Soil microbes in the Arctic tundra could survive at temperatures below $-7$ to $-11\,°C$ (Lipson et al., 2000; Männistö et al., 2009; Lipson and Monson, 1998), the general threshold for microbial cell lysis in non-tundra environments (D. Gao et al., 2018, 2021; Song et al., 2017). The microbes that can survive under these freezing conditions actively play a role in decomposing available DOC in the surface organic layer during thaw phases in FTCs. In addition, the top organic layer was composed of a higher quality plant-derived organic matter compared to the underlying mineral layer in the moist acidic tundra in Council, Alaska, which is the same as our study site (White et al., 2004). Thus,

the biologically labile DOC could be available in the surface organic layer (L. Gao et al., 2018). Hence, decreases in DOC associated with activated microbial activities following FTCs suggest that responses of the DOC in the organic layer to FTCs would be crucial in affecting the tundra C cycle under Arctic warming. More frequent FTCs and a longer thawing length in tundra soils with warming could enhance soil C availability in the active layer of the Arctic terrestrial ecosystems, leading to a high risk of $CO_2$ being released into the atmosphere (Estop-Aragonés et al., 2020).

Meanwhile, FTCs did not significantly change the activities of extracellular enzymes (Table 2), which are released by soil microbes to obtain C and N from recalcitrant soil organic matter such as cellulose, chitin, polypeptides, and lignin (Sinsabaugh, 2010; Liao et al., 2022). However, the enzyme activities in this study were measured under laboratory conditions with sufficient substrate supplies and suitable environment; therefore, these potential activities may not properly reflect actual microbial enzyme activities under the field conditions. Despite this inherent limitation, we argue that there is a non-significance in measured enzyme activity caused by FTCs, as soil microbes preferentially utilize simple compounds that do not require enzymes for degradation in DOC decomposition enhanced by FTCs (Foster et al., 2016; Gao et al., 2021; Perez-Mon et al., 2020). These re-

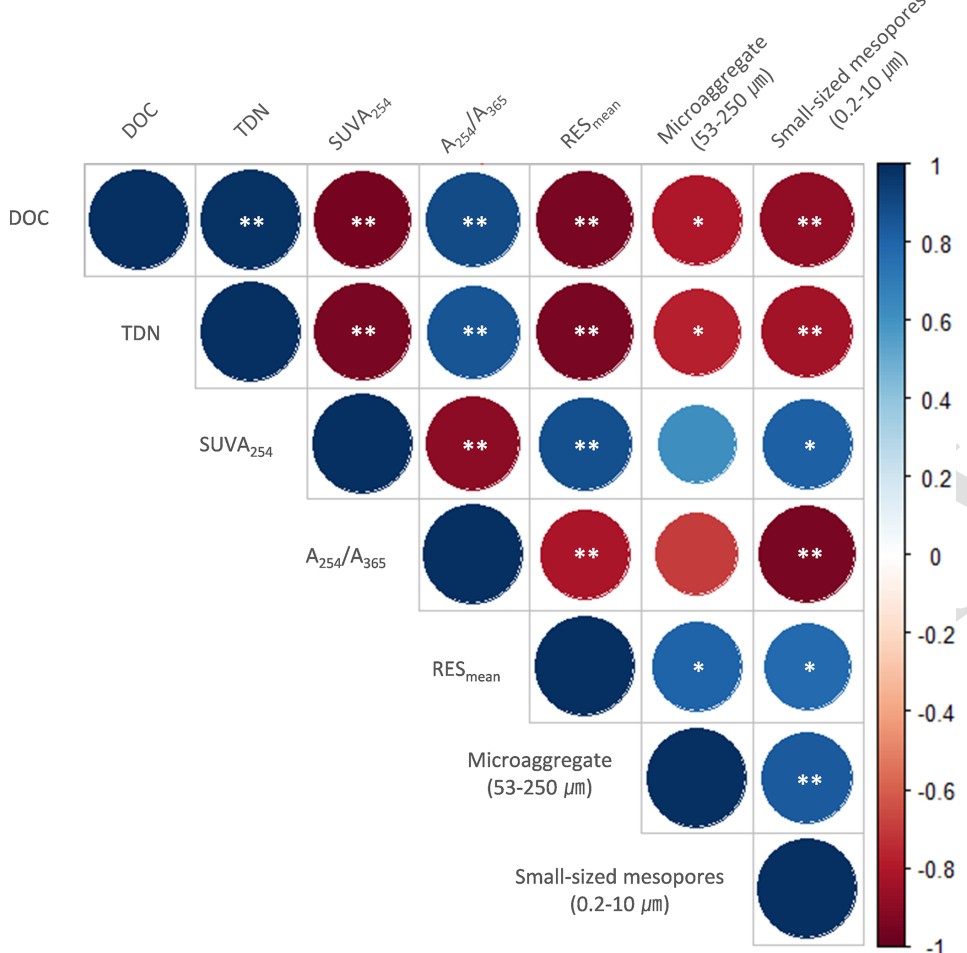

**Figure 3.** Correlation matrix between the observed variables in the FTC and CON soils. The analysis was performed on the observed variables that showed significant differences between treatments. Cool (with maximum blue) and warm (with maximum red) colors represent positive and negative correlations, respectively. The asterisks ** and * indicate significant correlations at the $p < 0.05$ and $p < 0.10$ levels, respectively.

sults were evidenced by the different DOC quality between the FTC and CON soils (Table 1). The DOC quality indices, $SUVA_{254}$ and $A_{254}$ / $A_{365}$, significantly differed between the FTC and CON soils, indicating that complex substrates with high aromaticity and molecular weight remained in the dissolved organic matter after successive FTCs (Berggren et al., 2018; Yang et al., 2019).

A series of multivariate analyses indicated the relationships between DOC characteristics and soil microbial activities influenced by FTCs. The results in the PCA and Pearson's correlation analysis showed that soil $CO_2$ production, influenced by FTCs, was closely related to the quantitative and qualitative changes in DOC (Figs. 2 and 3). Furthermore, as shown in Table 5, MLR analyses identified that those relationships are direct. These results eventually confirmed the first hypothesis that FTCs can change DOC quantity and quality without inhibiting soil microbial activities previously adapted to temperature fluctuations in the Arctic.

## 4.2 Effects of freeze–thaw cycles on dissolved organic carbon associated with soil structural properties

FTCs caused an increase in microaggregates (53–250 μm) and a corresponding decrease in mineral-associated fractions ($< 53$ μm), despite low significance levels (Table 3). In other words, the formation of microaggregates by FTCs is likely enhanced by the binding of smaller-sized aggregates rather than the breakdown of larger-sized ones. This could be related to the ice formation in the soil as the ambient temperature drops to $-9.0 \pm 0.3\,^\circ$C during the FTCs (Fig. S1). Under the freezing phase of FTCs, soil water gradually freezes, but at a microscale level, it can still form thin films of unfrozen water on the surfaces of soil particles. These unfrozen water films can intensively contain dissolved solutes that are charged or excluded during icing, which contribute to stabilizing soil structure (Sletten, 1988; Zhang et al., 2016). These characteristics allow the unfrozen water films to func-

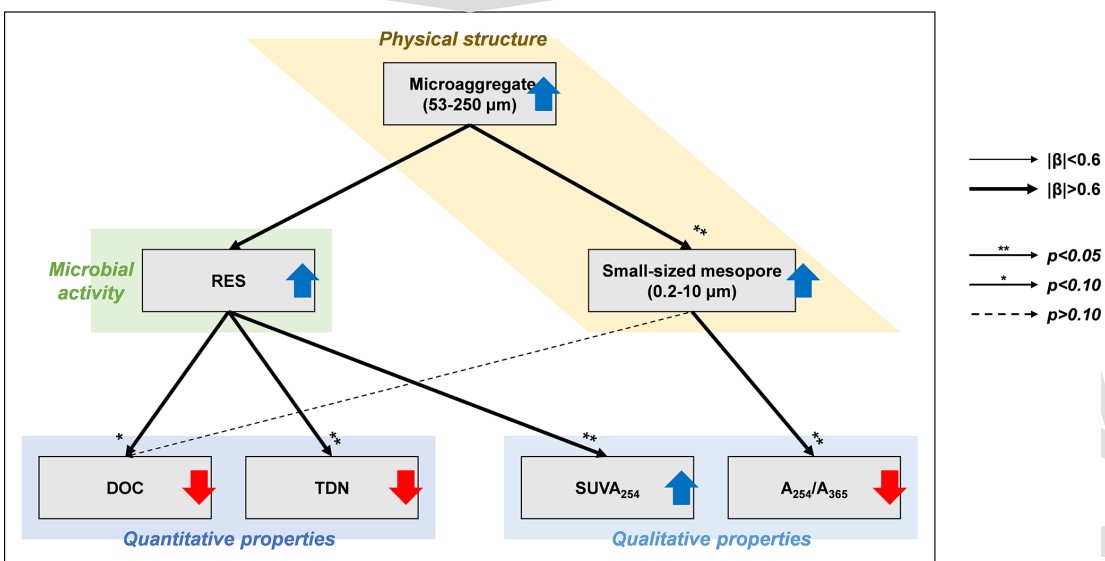

**Figure 4.** Conceptual diagram illustrating the response of surface organic soils in the Arctic tundra to the freeze–thaw cycles (FTCs). The upward blue and downward red arrows represent the increase and decrease in the observed variables by FTCs, respectively. The correlation strength and significance were determined through multiple linear regression (MLR) analyses (Table 5). The line widths depict the magnitude of the standardized $\beta$ coefficients. The arrows with asterisks indicate significant correlations at the $p < 0.05$ and $p < 0.10$ levels, respectively, while the dashed arrow without an asterisk indicates a correlation at a $p > 0.10$ level.

tion as binding agents between soil particles, enhancing soil microaggregation after seven successive FTCs. This is further supported by the fact that the extent of the decrease in mineral-associated fractions, despite no significance, is comparable to that of the increase in microaggregates (Table 3).

Soil microaggregates enhanced by FTCs affected DOC quantity and quality mainly through changes in the microbe-mediated mechanism rather than the direct pathway (Fig. 4). Because soil structural dynamics derived from FTCs can be a critical process for soil quality and function (Rabot et al., 2018), soil aggregate formation can improve soil structural stability and govern nutrient cycling and water retention, resulting in enhanced microbial activity (Bird et al., 2000; Yoo et al., 2017; Kim et al., 2021). Consequently, our findings suggest that soil structural improvement by FTCs, at the microaggregate scale, contributes to DOC decomposition by soil microbes, thereby resulting in reduced DOC content and increased DOC aromaticity in the FTC soil (Fig. 4).

Furthermore, the quantitative and qualitative changes in DOC can be attributed to the formation of specific-sized pores by soil microaggregates enhanced by FTCs. We found that the significant difference in soil PSD affected by FTCs was in the small-sized mesopores (Table 5). These pores were strongly related to soil microaggregate formation (Figs. 3 and 5). Our findings indicate that the increase in microaggregate formation by FTCs likely created the corresponding small-sized mesopores through the rearrangement

and formation of soil pores (Peng et al., 2015; Dal Ferro et al., 2012; Zaffar and Lu, 2015). Furthermore, such pores are able to hold water surrounding the soil particles (Jim and Ng, 2018; Kim et al., 2021), potentially contributing to water-film development on the soil particle surfaces. As previously mentioned, these water films can have electrical charges and condensed solutes during the freezing periods of the FTCs, serving as binding materials for soil microaggregation (Zhang et al., 2016). Thus, increase in mesopores by enhanced soil microaggregation may permit the dissolved solutes in the soil pore water to be adsorbed and occluded to the soils, thereby decreasing the DOC content in the soil solution of the FTC treatment (Fig. 4).

Multivariate analyses confirmed the second hypothesis about the quantitative and qualitative characteristics of DOC associated with soil structural changes by FTCs. The FTCs led to an increase in soil microaggregate formation and consequent changes in soil microbial activity and pore distribution, accelerating DOC decomposition and decreasing its content in the soil solution (Figs. 2 and 3; Table 5). Our findings contribute to a mechanism-based understanding of the effect of FTCs on DOC properties through systematic measurements on soil biogeochemical properties.

## 5 Conclusions

This study demonstrated the responses of organic soils in the Arctic tundra to FTCs, focusing on the changes in DOC characteristics associated with microbial activity and soil physical structure. We found that the following seven variables differed significantly after FTCs: soil $CO_2$ production ($CO_2$), DOC and TDN contents, two DOC quality indices ($SUVA_{254}$ and $A_{365} / A_{254}$), microaggregate (53–250 µm) distribution, and small-sized mesopore (0.2–10 µm) proportion. Multivariate statistical analyses, including PCA, Pearson's correlation, and MLR, contributed to the mechanism-based interpretation of how FTCs altered DOC quantity and quality mediated by the changes in microbial activity and soil physical structure. As a result, FTCs altered the DOC quantity and quality with higher $CO_2$, indicating that FTCs affected DOC characteristics without negatively impacting microbial activity. In addition, soil microaggregation enhanced by FTCs and the subsequent increase in soil $CO_2$ production and small-sized pore distribution could promote DOC decomposition, eventually decreasing the DOC content in the soil solution. In conclusion, we elucidated the effects of FTCs on DOC characteristics in the Arctic organic soils of active layer by incorporating soil structural changes and microbial responses. Further study is required to determine how the deeper active layer or ice-rich permafrost thaw under warming would affect the permafrost C dynamics with FTCs.

*Data availability.* All data can be provided by the corresponding author upon request.

*Supplement.* The supplement related to this article is available online at: https://doi.org/10.5194/tc-17-1-2023-supplement.

*Author contributions.* YJK and JYJ planned the campaign; YJK and JK performed the measurement and analyzed the data; YJK wrote the manuscript draft; JYJ and JK reviewed and edited the paper; JYJ acquired the financial support for the project leading to this publication.

*Competing interests.* The contact author has declared that none of the authors has any competing interests.

*Acknowledgements.* This study was supported by the National Research Foundation of Korea funded by the Korean Government (grant no. NRF-2021M1A5A1075508/KOPRI-PN23012). We are grateful to Sungjin Nam, a researcher at the Korea Polar Research Institute, for performing fieldwork and soil sampling.

*Financial support.* This research has been supported by the National Research Foundation of Korea (grant no. NRF-2021M1A5A1075508/KOPRI-PN23012).

*Review statement.* This paper was edited by Hanna Lee and reviewed by Liam Heffernan and one anonymous referee.

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

**Remarks from the typesetter**

TS1    Please give an explanation of why this needs to be changed. We have to ask the handling editor for approval. Thanks.

TS2    Please give an explanation of why this needs to be changed. We have to ask the handling editor for approval. Thanks.

TS3    Please note that it is our standard to repeat the percent sign in all instances.

TS4    Please confirm as the last access date can also be today's date if the link is still active.