# Peer review of "Responses of dissolved organic carbon to freeze–thaw cycles associated with the changes in microbial activity and soil structure You Jin Kim, Jinhyun Kim, and Ji Young Jung"

_The Cryosphere, 2023_

## Referee Comment (RC2)

[referee-annotated manuscript omitted]

---

## Author Comment (AC1)

**Reviewer #1**

1. It is clear from context clues throughout the paper that the authors sampled active layer soil. However, this needs to be more clear. Please provide the depths of sampling (or, if I missed them somewhere, please make them more visible/clear). Overall, the introduction and discussion both do a nice job of setting up and discussing the use of active layer soil, but it just needs to be explicitly stated.

   ⇨ According to the reviewer's comments, we further explained and clarified our experimental design and the overall description for soil sampling (Sections 2.1).

   ⇨ Revised as:

   > **2.1. Site description and soil preparation**
   >
   > Soil samples for microcosm incubation were collected from the moist acidic tundra in Council (64.51° N, 163.39° W) on the Seward Peninsular in Northwest Alaska. The average temperature and precipitation over the past 30 years (1981–2020) are -3.1 °C and 258 mm (Alaska Climate Research Center). In the early spring (April to May) the minimum and maximum temperatures are -8.5 and 7.1 °C, respectively (Alaska Climate Research Center). This site is a tussock tundra dominated by cotton grasses (*Eriophrum vaginatum*), blueberries (*Vaccinium uliginosum*), lichen, and moss (*Sphagnum* spp.).
   >
   > Soil sampling was performed at three random points under similar vegetation compositions. Each point was within approximately 100 m distance from each other. At the time of sampling (early July 2010), the active layer depth was approximately 50 cm, measured by a steel rod (1 m). Soil samples were acquired by hammering a stainless steel pipe (7.6 cm diameter × 50 cm long) into the partially- or well-degraded organic layer (Oe), mixed with soil minerals, after removing the litter layer (Oi) on the surface. The soil samples were stored at -20 °C before initiating microcosm incubation. The frozen soil was thawed at <4 °C, and the surface organic soils were passed through a 2-mm sieve and homogenized by hand. Fig. 1 shows the soil sampling and preparation procedure. Soil textural analysis was conducted by a wet sieving and pipette method (Kim et al., 2022). Soil bulk density (BD) was determined by calculating the soil dry weight contained in the soil core volume. Volumetric water content (VWC) in the soil was measured using a portable sensor with an accuracy of ±0.01 $cm^3$ $cm^{-3}$ (Procheck Daegon Devices, Washington, US). Total carbon (C) and nitrogen (N) contents were determined through combustion (950 °C) with an elemental analyzer (vario MAX cube; Elementa varioMAX cube; Elementar, Langenselbold, Germany). The basic soil properties are summarized in Table S1.

2. Why were organic soils sampled, rather than mineral? Apologies if I missed this somewhere. It seems like it would have made more sense to do PSD and aggregate stability analysis on mineral soils. Or both organic and mineral. Could you add a sentence defending the focus on organic soils rather than mineral? I believe some of the papers you are citing focused on mineral deformation by freeze-thaw cycles, but please correct me if I'm wrong.

   ⇨ We focused on the topsoil layer, i.e., the organic layer in this study site (acidic moist tundra), since the surface soil can be firstly and directly affected by the FTCs. In [L35-40], we described the relevant contents in the Introduction for the justification of our experimental materials: "Moreover, increased temperature in Arctic regions accelerate snow melting (Henry, 2008; Førland et al., 2011; Kreyling et al., 2008) and cause rainfall instead of snowfall (Henry, 2013; IPCC, 2014), leading to the absence of snow cover on the soil surface (Callaghan et al., 1998; Heal et al., 1998). In Arctic regions, snow plays a key role in protecting tundra soils against dramatic temperature changes caused by harsh climates (Royer et al., 2021). Exposed soil surfaces lacking snow cover are likely to undergo more frequent freeze-thaw cycles (FTCs) in the early spring and late autumn because they are directly influenced by the diurnal fluctuations of atmospheric temperature (Kreyling et al., 2008; Henry, 2013; Freppaz et al., 2007)."

   ⇨ As the reviewer said, the soil aggregate fractionation and pore size distribution (PSD) mostly have been analyzed in mineral soil. This is because the arrangement and assemblage of primary mineral particles such as sand, silt, and clay would mainly determine those properties. However, there are also results showing PSD and soil aggregate fractionation using diverse types of soil types under high organic matter contents, such as grassland, peatlands, bogs, fens, and marshes (Garcia-Franco et al., 2021; Pu et al., 2022; Weber et al., 2016; Dettmann et al., 2014). The above mentioned studies showed that organic-rich soils containing some mineral particles can be used for examination of soil aggregates and pores structures. In our experiment, the soil for the incubation was collected from the Oe layer, where organic and mineral particles are well mixed, excluding the Oi layer mainly composed of less-decomposed plant leaves or roots. This could enable us to analyze soil aggregate fractionation and PSD with the sampled soil. In [L86-88], soil preparation was described in detail to show that our soil samples could be utilized for analyses of soil aggregate fractionation and PSD: "Soil samples were acquired by hammering a stainless steel pipe (7.6 cm diameter × 50 cm long) into the partially- or well-degraded organic layer (Oe), mixed with soil minerals, after removing the litter layer (Oi) on the surface."

3. Personally, I would like to see a diagram of each of the three cores collected via SIPRE corer with depths of organic and mineral soils as well as active layer and permafrost shown. I would also be interested in knowing how much of the organic soil you subsampled for the experiment. Did you include peat layers? Or just the more decomposed muck?
   ⇨ To improve comprehensibility of our experimental design, we added a new figure (Fig. 1) suggested by the reviewer.

[Figure]

**Figure 1: Soil sampling and experimental design for the microcosm incubation study**

   ⇨ In addition, we added information on soil conditions at the time of sample acquisition in [L86-88]: "Soil samples were acquired by hammering a stainless steel pipe (7.6 cm diameter × 50 cm long) into the partially- or well-degraded organic layer (Oe), mixed with soil minerals, after removing the litter layer (Oi) on the surface."

4. I agree that 12 hours is enough to freeze and thaw 120 g of soil, but please cite another incubation for the method or provide test data where you found that soil was able to completely freeze and thaw in that time. It will make your method more citable/reproducible.
   ⇨ Our experimental set-up should ensure the complete freezing and thawing of the soil under the temperature fluctuations of -9~6℃ at 12-hr intervals for the reliability of the experiment as the reviewer's comment. Thus, we cited several more studies applied with the similar experimental set-up to our FTCs simulations and reported significant differences in soil properties. In [L106-108], added as: "We ensured that our FTC treatment was adequate for complete freezing and thawing of the soil based on previous studies conducted under similar conditions (Freppaz et al., 2007; Larsen et al., 2002; Song et al., 2017; Han et al., 2018)."

5. I am unclear on the significance of 7 freeze-thaw cycles in the context of your paper. Ma et al., 2021 found unpredictable freeze-thaw response up to 7 freeze-thaw cycles, after which freeze-thaw resulted in increased pore connectivity. Placing your experiment on this cusp, at 7 freeze-thaw cycles, is a great contribution, but I think it needs to be a little bit more clear why you chose 7 freeze-thaw cycles. Was it because you were targeting that unpredictable pore network response that ends at around 7 freeze-thaw cycles? (line 94)
   ⇨ We thought that seven-successive FTCs might be sufficient to anticipate soil biogeochemical changes induced by freeze-thaw events. Previous studies have shown that soil carbon dynamics (Gao et al., 2021) and total porosity (Liu et al., 2021; Ma et al., 2021) can be responded to the 3-10 FTCs. In [102-104], we added the references for why we adopted 7 FTCs: The meta-analysis and several other studies showed that soil carbon dynamics (Gao et al., 2021) and total porosity (Liu et al., 2021; Ma et al., 2021) responded to 3–10 FTCs; thus, seven-successive FTCs were adopted in this study.

6. I would like a little more information about the soil processing pre-incubation. No sieving occurred? Existing aggregate structure was maintained from coring to subsampling to incubation? (lines 80-87)
   ⇨ Admitting the reviewer's point, we added more information about soil preparation. "The soil samples were stored at -20 ℃ before initiating microcosm incubation. The frozen soil was thawed at <4 ℃, and the surface organic soils were passed through a 2-mm sieve and homogenized by hand. [L88-89]."

⇨ In addition, we were aware that the existing aggregate structure of the in-site soil could be disturbed by sieving and homogenization. Nonetheless, these processes were necessary for comparing the changes in the FTC and CON soils under same initial conditions.

7. Can you expand on what you mean by "adjusting soil bulk densities" (line 98) How was this done? Do you just mean you packed the sample containers with a measured mass of soil to achieve a known bulk density?

⇨ Yes, we adjusted the bulk density as the reviewer understood. Since we used disturbed soils rather than intact core samples for incubation, we tried to make the initial soil conditions as uniform as possible. We added a detail description how we adjusted the conditions for incubating soils similar to field status. In [L109-113], added as: "In all incubation sets, the initial soil BD was adjusted to 0.72 g cm$^{-3}$, similar to field-soil conditions (Fig. 1; Table S1). For the destructive sampling set, we used 260 g of homogenized fresh soil (154.7 g dry weight) to a soil volume of 215 cm$^3$. The microcosm set using the small-sized cores was established with 120 g of homogenized fresh soil (71.5 g dry weight) in a 99 cm$^3$ volume. The VWC for all incubation soils was also standardized by spraying water using a pipette to 0.50 cm$^3$ cm$^{-3}$ (70% water-filled pore space), a similar level to field soils (Fig. 1; Table S1)."

8. Please be clear in the discussion when you are comparing findings from a paper that looked at mineral soil with your organic soil-based experiment. It's okay to have both, but I think it's important to be really clear if there are contrasting material types. I thought Ma et al., 2019; 2021 and Liu et al., 2021 generally looked at mineral soil but I could be wrong.

⇨ According to the reviewer's comment, we reviewed our references. As a result, previous studies other than research results focusing on organic soils were removed from Introduction and Discussion (such as Ma et al., 2019; 2021, Liu et al., 2021).

9. Lines 223-224: I understand that speculation, but I would point out that in Rooney et al., 2022, changes in pore structure were observed at both low and high water contents. So that speculation could be an oversimplification of how the geometry and architecture of the pore network or even the direction of the freezing front influence pore deformation during freeze-thaw.

10. Lines 223-224: Do you expect that much evapotranspiration to have taken place? It wasn't clear to me in your methods that water was continously evaporating throughout the experiment. Could you specify if the samples were generally kept sealed from the atmosphere in some way to prevent evaporation?

⇨ (Responses to comments 9 and 10) Admitting the reviewer's point, we have deleted the relevant sentences. We acknowledge that our experimental design cannot show the effect FTCs on soil body or aggregate breakdown under different soil moisture conditions. Therefore, we focused more on the explanation of micro-aggregate enhanced by FTCs.

11. Figures 1-2: I really like both of these figures.

⇨ Thank you for your compliment.

12. Figure 3: I think the figure caption could have a little bit more information in it. Specifically could you explain the legend? For a couple minutes I thought the colors had something to do with temperature, especially with the placement of the color bar in the figure.

⇨ According to the reviewer's comment, we added more information for the figure.

[Figure]

Figure 1: Correlation matrix between the observed variables in the FTC and CON soils. The analysis was performed on the observed variables that showed significant differences between treatments. Cool (with maximum blue) and warm (with maximum red) colors represent positive and negative correlations, respectively. The asterisks ** and * indicate significant correlations at the $p<0.05$ and $p<0.10$ levels, respectively.

13. Figure 4: I hope someone else can provide feedback to you on this. I am unfamiliar with structural equation model analysis.
   ⇨ As per the feedback to the second reviewer's comment, considering the size and characteristics of the data, we have represented the soil structural and biogeochemical responses to FTCs utilizing multiple linear regression (MLR) analyses (Fig. 4; Table 5), instead of a structural equation model (SEM). Accordingly, the figure depicting SEM has been removed.

14. I was disappointed to not see PSD plots (y axis = pore volume, x axis = pore diameter). Maybe the plots could be included in supplemental? Personally, I would like to see a PSD graph for each individual sample to get an idea of overall sample heterogeneity, rather than just the standard error for each size class. Showing sample heterogeneity seems especially important in organic soils.
   ⇨ According to the reviewer's point, we added the soil water release curves and PSD plots in supplementary materials (Fig. S2).

[Figure]

**Figure S2. Water release curves and pore size distribution in the FTC and CON soils.**

15. Check for typos throughout, although I didn't see many.
   ⇨ We got help from the professional English editing service before the first submission, and again did with this revised MS according to the reviewer's recommendation, to check for any typos and grammatical errors.

16. This is a very exciting and cool paper! Looking forward to the published version.
   ⇨ We appreciate your thoughtful feedback. We believed that your critical comments and questions greatly improved our MS. We have carefully crafted responses and reflected revisions, so hopefully, our revised manuscript adequately addresses your concerns.

[revised manuscript text omitted]

---

## Author Comment (AC2)

**Reviewer #2**

**General comments**

My recommendation for this manuscript is to resubmit following major revisions. While the results are of high interest to The Cryosphere, the manuscript is currently lacking in information to be considered for immediate publication. I's current form, it is unclear exactly how the authors conducted the experiment, and it would not be possible to repeat this study given the information provided. The authors have also not fully described, presented, and explained their results. Both the methods and results section of this manuscript can be greatly improved with more information.

⇨ We sincerely value your feedback provided on the manuscript. We have made extensive revisions to address your comments in a line-by-line manner thoroughly. We hope that our responses have adequately addressed your comments and critics. We are grateful for your feedback and the opportunity to improve our manuscript.

One issue I have currently with the interpretation of the results is the presentation and conclusions drawn from the structural equation model (Figure 4). I suggest the authors reconsider how this is presented, provide a detailed a prior model that includes the hypothesis and justification behind each pathway, and the soundness of each pathway.

⇨ Before addressing the reviewer's comments, we reconfirmed whether our data set is the appropriate size and characteristics for applying SEM. As one of the fundamental requirements for SEM is to have a sufficient sample size (N) of at least 5-10 times the number of estimated parameters, typically exceeding 150 cases, we admitted that our data structure was unsuitable for SEM. Instead, we adopted the multiple linear regression (MLR) analyses, a statistical technique with a lower complexity level than SEM and seriously re-considered the pathway as well. Thus, we have revised the statistical methods and modified the results throughout the manuscript.
* * *
**Abstract**

[L13-15] "Multivariate statistical analyses supported that the FTCs improved soil structure and functions which led to facilitated DOC decomposition by soil microbes, and changes in DOC quantity and quality by FTCs."

- "…soil structure…" How do they assess this?
  ⇨ In this study, we examined soil structure at the level of aggregates and pore distribution rather than a larger-scale of soil bodies. To avoid confusion, we specifically described the scale of soil structures we targeted. In [L14-15], revised as: "Multivariate statistical analyses indicated that the FTCs improved soil structure at the scale of micro-aggregates and small-sized mesopores, facilitating DOC decomposition by soil microbes and changes in DOC quantity and quality by FTCs."

- "…led to…" Remove
  ⇨ Admitting the reviewer's point, we have removed it.
* * *
**Introduction**

[L26-28] "Recently, Arctic warming, four times faster than global warming (IPCC, 2019), have enhanced permafrost thaw, causing the previously stored SOC to be released into greenhouse gases ($CO_2$ and $CH_4$) and/or leaching dissolved organic carbon (DOC) (Estop-Aragonés et al., 2020)."

- "IPCC, 2019" Correct reference? Was this included in the IPCC report?
  ⇨ We are grateful for the reviewer's careful review. According to the reviewer's comment, the reference has been replaced with a new one corresponding to the content [L26].
  ⇨ Reference:
    Rantanen et al. The Arctic has warmed nearly four times faster than the globe since 1979. Commun Earth Environ 3, 168 (2022). https://doi.org/10.1038/s43247-022-00498-3.

- "…have…" → has
  ⇨ Admitting the reviewer's point, we have revised it [L26].

[L31] "Permafrost thaw also influences the Arctic watershed by inflowing terrestrial-derived DOC into the surrounding lakes and seas…" → export of
  ⇨ In [L30], revised as: "…the Arctic watershed by export of terrestrial-derived DOC…"

[L31] "The released DOC in the active layer…" → exported? mobilized? transported?
  ⇨ In [L31], revised as: "The exported DOC from the active layer…"

[L34] "Thus, the measurement for quantitative and qualitative DOC changes…" → Of
  ⇨ Admitting the reviewer's point, we have revised it [L33].

[L36] "Moreover, increased temperature of Arctic regions…" → in
  ⇨ Admitting the reviewer's point, we have revised it [L35].

[L44-45] "Numerous studies have reported influences of FTCs on soil C availability, which is strongly related to $CO_2$ emissions and microbial growth/activity in Arctic tundra soils..." this reads like they soil carbon availability is driven by $CO_2$ respiration and microbial growth, rather than the other way around.
  ⇨ We admitted that our writing could cause the confusion to the reader. We have revised the sentence to clarify this point as "Numerous studies have reported influences of FTCs on the labile soil C content, which is strongly related to microbial activity in Arctic tundra soils…" in [L43-44].

[L47-49] "In surface soil, FTCs could increase the amount of DOC in soil solution… (Sawicka et al., 2010; Schimel and Clein, 1996; Larsen et al., 2002)." Are these tundra soils studies?

⇨ Upon reviewing our references further, we have found that all of the studies, except for one by Sawicka et al. (2010), are related to the Arctic topsoil. In [L46-48], revised as: "FTCs have been reported to increase the amount of DOC in a few tundra and non-tundra soils, attributed to a decrease in microbial utilization of DOC due to cell lysis generally occurring below -7 to -11 °C of freezing temperature (Gao et al., 2018a, 2021; Song et al., 2017; Schimel and Clein, 1996; Larsen et al., 2002)."

[L51] "This was interpreted by that soil microorganisms in the Arctic permafrost…" → as

⇨ Admitting the reviewer's point, we have revised it [L51].

[L76-77] "We tested the following hypotheses: …and (2) change in soil micro-aggregation by FTCs enhance microbial activities and water-holding pores, eventually affecting DOC characteristics." in what way?

⇨ We revised it to clarify our hypotheses. In [L74-76], revised as: "We tested the following hypotheses: …(2) soil aggregate distribution influenced by FTCs changes DOC characteristics by enhancing microbial activities and altering specific-sized soil pore proportion."

**Materials and methods**

[L78] 2. Materials and methods: Overall, the information provided in the methods section is limited and this experiment would be difficult to replicate based on the information provided. Sections 2.1 and 2.2 in particular need more information and clarity on experimental design, what analysis was done and how it was done, and whether this effects the soil parameters in situ or solely in the experimental set up.

⇨ We agreed with the reviewer's comments. To further explain and clarify the experimental design, we improved the overall description of soil sampling and experimental design (Sections 2.1 and 2.2) and added a new figure (Fig. 1).

[Figure]

**Figure 1: Soil sampling and experimental design for the microcosm incubation study**

[L79] "2.1. Soil preparation" This section does not deal with soil preparation, rather it is a site description.

⇨ Admitting the reviewer's point, we changed the title of Section 2.1 to "Site description and soil preparation." Furthermore, this section was revised to describe the contents in detail.

⇨ Revised as:

**2.1. Site description and soil preparation**

Soil samples for microcosm incubation were collected from the moist acidic tundra in Council (64.51° N, 163.39° W) on the Seward Peninsular in Northwest Alaska. The average temperature and precipitation over the past 30 years (1981–2020) are -3.1 °C and 258 mm (Alaska Climate Research Center). In the early spring (April to May) the minimum and maximum temperatures are -8.5 and 7.1 °C, respectively (Alaska Climate Research Center). This site is a tussock tundra dominated by cotton grasses (*Eriophrum vaginatum*), blueberries (*Vaccinium uliginosum*), lichen, and moss (*Sphagnum* spp.).

Soil sampling was performed at three random points under similar vegetation compositions. Each point was within approximately 100 m distance from each other. At the time of sampling (early July 2010), the active layer depth was approximately 50 cm, measured by a steel rod (1 m). Soil samples were acquired by hammering a stainless steel pipe (7.6 cm diameter × 50 cm long) into the partially- or well-degraded organic layer (Oe), mixed with soil minerals, after removing the litter layer (Oi) on the surface. The soil samples were stored at -20 °C before initiating microcosm incubation. The frozen soil was thawed at <4 °C, and the surface organic soils were passed through a 2-mm sieve and homogenized by hand. Fig. 1 shows the soil sampling and preparation procedure. Soil textural analysis was conducted by a wet sieving and pipette method (Kim et al., 2022). Soil bulk density (BD) was determined by calculating the soil dry weight contained in the soil core volume. Volumetric water content (VWC) in the soil was measured using a portable sensor with an accuracy of ±0.01 $cm^3$ $cm^{-3}$ (Procheck Daegon Devices, Washington, US). Total carbon (C) and nitrogen (N) contents were determined through combustion (950 °C) with an elemental analyzer (vario MAX cube; Elementa varioMAX cube; Elementar, Langenselbold, Germany). The basic soil properties are summarized in Table S1.

[L83-85] "Soil sampling was performed at three random locations chosen as replicates using a soil core sampler (SIPRI corer, John's Machine Shop, Fairbanks, AK, USA)…"

- "…three random locations…" how far apart? do they have similar vegetation? can they be considered as replicates of the site? How do you justify that they are adequate site replicates?
  - ⇨ To answer the reviewer's point and justify the acquired soil samples as replicates, we revised the description of soil sampling points. In [L84-85], revised as: "Soil sampling was performed at three random points under similar vegetation compositions. Each point was within approximately 100 m distance from each other."
- "…soil core sampler…" what depth were the cores taken to? Do you include surface organic and vegetation layer? Were the soils frozen when you took the core? How many soil cores from each location did you take?
  - ⇨ To answer the reviewer's point, we revised and added the description of soil sampling. In [L85-88], revised as: "At the time of sampling (early July 2010), the active layer depth was approximately 50 cm, measured by a steel rod (1 m). Soil samples were acquired by hammering a stainless steel pipe (7.6 cm diameter × 50 cm long) into the partially- or well-degraded organic layer (Oe), mixed with soil minerals, after removing the litter layer (Oi) on the surface. The soil samples were stored at -20℃ before initiating microcosm incubation."
- "SIPRI corer" Should be SIPRE (Snow, Ice, Permafrost Research Establishment) corer?
  - ⇨ Sorry for the confusion. Soil was collected using a stainless steel pipe (7.6 cm diameter × 50 cm long), not a SIPRE corer. Thus, we revised it.

[L85-87] "The soil core samples were divided into organic and mineral layers, of which the surface organic soils were collected and homogenized for basic soil analyses and incubation experiments."

- "…organic and mineral layers…" what depths did these correspond to? how did you divided them? was it based on a visual inspection? how much soil did you actually collect from the organic layer?
  - ⇨ We presented Figure 1 with information on the depth of the organic and mineral layers, including the acquired soil core samples. The separation of organic soils from the cores was conducted based on a visual inspection and our previous data on soil carbon content (>20%).
- "…homogenized…" how were the homogenized? By hand? sieving?
  - ⇨ To explain the reviewer's point, we revised the sentences in [L88-89]: "The frozen soil was thawed at <4℃, and the surface organic soils were passed through a 2-mm sieve and homogenized by hand."
- "…basic soil analyzes…" please describe what these involve and how you measured them. How was soil texture determined? Bulk density and volumetric water content?
  - ⇨ We added the analytical methods of soil properties in [L90-95]: "Soil textural analysis was conducted by a wet sieving and pipette method (Kim et al., 2022). Soil bulk density (BD) was determined by calculating the soil dry weight contained in the soil core volume. Volumetric water content (VWC) in the soil was measured using a portable sensor with an accuracy of ±0.01 $cm^3$ $cm^{-3}$ (Procheck Daegon Devices, Washington, US). Total carbon (C) and nitrogen (N) contents were determined through combustion (950℃) with an elemental analyzer (vario MAX cube; Elementa varioMAX cube; Elementar, Langenselbold, Germany)."

[L89-90] "Soil incubation was conducted with two parallel sets of microcosms: one for destructive sampling and the other for monitoring soil PSD changes. For the destructive sampling set, 260 g of fresh soil in a 380 mL polypropylene bottle was used…"

- "…two parallel sets of microcosms…" were these run in replicate based on site replicates? so each incubation had 3 microcosms?
  - ⇨ Yes, each microcosm had three replicates, respectively. We added a new figure (Fig. 1) to help understanding of the experimental design.
- "…polypropylene bottle…" was the bottle pre-treated before? Acid washed?
  - ⇨ Before transferring soil, the bottle was sterilized with alcohol, washed with deionized water, and oven-dried.

[L91-93] "The other incubation set used a re-constructed soil core by placing 120 g of fresh soil in a 99 mL core cylinder (5 cm diameter × 5 cm height) to measure PSD alterations under FTCs conditions."

- "…120 g of fresh soil…" this is the homogenized soil? how was the soil homogenized? how does placing fresh soil into a cylinder impact the soil structure and is this meant to represent similar soil structure to what is seen in the field? Or does this type of design result in an artificial soil structure that you then manipulate through FTCs? If this, then how do you justify that this is similar to what may occur in situ? How does the PSD in your experiment compare to PSD in situ?
  - ⇨ We understood that the reconstructed soil is not field-like due to the disturbances before incubation experiments. However, this experiment focused on changes in FTC soils compared to the control after incubation rather than examining differences in soil conditions between the field and lab-incubation. Therefore, this experiment cannot perfectly imitate the effects of FTCs on soil structure in situ since the reconstructed soil cores are not same as the field condition (we could not be simulated the field conditions for the preferential flows such as cracks, macropores roots, and so on with the reconstructed soil cores). Nevertheless, the standardized initial conditions could enable us to explore the mechanisms for the effects of FTCs. We did not compare the PSD in-situ with the reconstructed soils since the field collected soil could have some cracks, roots pores, etc. or be horizontally broken during samplings.
- "…99 mL core cylinder…" do you have a picture of this that could be included in a figure?
  - ⇨ Yes, we presented it in Fig. 1.
- "…PSD alterations under FTCs…" compared to what? in situ pore size, or artificial pore size distribution associated with this experimental design? I am not trying to argue that one is better that the other, but do think the authors should make this distinction clear and acknowledge this in their interpretation of the results.
  - ⇨ Admitting the reviewer's comment, we clarify the purpose of the second incubation design. As responded earlier, we focused on changes in FTC soils compared to the control after incubation rather than differences between field and incubation conditions.
  - ⇨ Thus, we revised the sentence in [L99-101]: "The other microcosm set was created by reconstructing the small-sized soil core (5 cm diameter × 5 cm long) to compare PSD alterations under incubation conditions with/without the impact of FTCs."

[L94-96] "This temperature range is representative of early spring conditions at the study site, where the average minimum and maximum temperatures from April to May are -8.5℃ and 7.1℃, respectively (Alaska Climate Research Center)." this could be included in section 2.1 where you describe the site conditions.
  - ⇨ Admitting the reviewer's point, we have moved it to Section 2.1.

[L97-99] "All incubation sets had soil bulk densities and volumetric water content adjusted to 0.72 g cm$^{-3}$ and 0.50 cm$^3$ cm$^{-3}$ (70% water-filled pore space), respectively, similar to field-soil conditions (Table 1)." so these were remeasured in the microcsosm? This is unclear. A more detailed description of the analysis procedure and on which samples these analyses were done is needed.
  - ⇨ We added a detail description how to adjust the bulk density and volumetric water content in incubation soils. In [L109-113], added as: "In all incubation sets, the initial soil BD was adjusted to 0.72 g cm$^{-3}$, similar to field-soil conditions (Fig. 1; Table S1). For the destructive sampling set, we used 260 g of homogenized fresh soil (154.7 g dry weight) to a soil volume of 215 cm$^3$. The microcosm set using the small-sized cores was established with 120 g of homogenized fresh soil (71.5 g dry weight) in a 99 cm$^3$ volume. The VWC for all incubation soils was also standardized by spraying water using a pipette to 0.50 cm$^3$ cm$^{-3}$ (70% water-filled pore space), a similar level to field soils (Fig. 1; Table S1)."

[L102-104] "Soil respiration was measured by collecting gas samples daily. The incubation bottle was sealed with a cap for 60 min, and a gas sample was collected from the headspace through a septum using 10 mL syringes (BD Luer-Lok tip, BD Company, Franklin Lakes, NJ, USA)."

- Was this done only during the FTC incubation? Were any respiration measurements made prior to beginning the incubations, or after?
  - ⇨ We measured soil respiration during the incubation period as a proxy for overall microbial activity in the soil; thus, it was measured throughout the incubation. To clarify the measurement period for soil respiration, revised as: [L116-118] "Soil respiration (RES), widely accepted as a proxy for overall microbial activity (Kim and Yoo, 2021; Maikhuri and Rao, 2012; Davidson et al., 1998; Kuzyakov and Domanski, 2000; Lipson and Schmidt, 2004; Raich and Schlesinger, 1992), was estimated by daily measurement of the CO$_2$ flux from soil incubation during the entire incubation period."
- just one sample? or multiple over the 60 minutes?
  - ⇨ We clarified the number of soil gas sampling and how to calculate the CO$_2$ concentration.
  - ⇨ In [L118-120], revised as: "We collected gas samples from the headspace through a septum using 10 mL syringes (BD Luer-Lok tip, BD Company, Franklin Lakes, NJ, USA) before and after sealing the incubation bottle for 60 mins."
  - ⇨ In [L125], revised as: "…where $dGas/dt$ is the change in the CO$_2$ concentration before and after sealing the incubation bottle for 60 mins,"

[L108-109] "…, where dGas/dt is the change in the CO$_2$ concentration over time, V and A are the volume and area of the incubation bottle, P is the atmospheric pressure, MW is the molecular weight of CO$_2$, R is a gas constant, and T is the absolute temperature." what values did you use for these? Was pressure set at 1? 0.96? what temperature did you use? include the gas constant and MW values.
  - ⇨ In [L126-127], added as: "…$P$ is the atmospheric pressure (1 atm), $MW$ is the molecular weight of CO$_2$ (44.01 g mol$^{-1}$), $R$ is a gas constant (0.082 atm L mol$^{-1}$ K$^{-1}$), and $T$ is the absolute temperature during gas collection (293 K)."

[L115-117] "Twenty grams of fresh soil was extracted with 40 mL of distilled water, filtered through a 0.45-μm filter, and measured using a Multi N/C 3100 analyzer (Analytik Jena, Jena, Thüringen, Germany)." How did this happen? Was the soil and water shaken for a period of time? were these placed in a centrifuge after? more detail needed.

⇨ We added the detail methods for measurement of DOC and TDN content through water extraction. In [L134-136], revised as: "After adding 40 mL of distilled water, 20 g of fresh soil were shaken for 1 h, centrifuged, and filtered with a 0.45-μm filter to obtain supernatant. The supernatants were measured using a Multi N/C 3100 analyzer (Analytik Jena, Jena, Thüringen, Germany)."

[L121-123] "For $NH_4^+$-N and $NO_3^-$-N contents analysis, 5 g of fresh soil was extracted using a 2 M KCl solution, and the filtrates were analyzed using an auto-analyzer (Quaatro, SEAL Analytical GmbH., Norderstedt, Schleswig-Holstein, Germany)."
- How was this done? what size pores were used on the filters?
⇨ We added the detail methods for measurement of $NH_4^+$-N and $NO_3^-$-N content through KCl extraction. In [L141-143], added as: "For $NH_4^+$-N and $NO_3^-$-N content analysis, 5 g of fresh soil was shaken with a 2 M KCl solution for 1 h, centrifuged, and filtered through Whatman #42 paper. The filtrates were analyzed using an auto-analyzer (Quaatro, SEAL Analytical GmbH., Norderstedt, Schleswig-Holstein, Germany)."

[L133-134] "To estimate the PSD, soil water release curves were generated by the Hydrus-1D model equipped with van Genuchten soilhydraulic equations, which can be applied to organic and mineral soil (Šimůnek et al., 2013)." on in situ soils or soils in mesocosms?
⇨ We revised the phrase to clarify the target soils for the PSD analysis. In [L153], revised as: "To estimate the PSD on the post-incubation core soils (5 cm diameter × 5 cm long)…"

**Results**

[L159] "3. Results" the results are very interesting and exciting, however they currently are not presented in a satisfactory way to show them off the best. They should be described and explored in more detail, particularly when discussing stat results. More information is required throughout. Also, the data from table 2 is nt correctly reported in the text. I do not know whether the text or the table is wrong, but this must be addressed as it is a key message of the paper. Do FTCs increase or reduce DOC and TDN concentrations?

⇨ We apologize for incomplete description in results. To clarify, we have thoroughly reviewed and confirmed the stat results and their descriptions. We clearly reported the reduced DOC and TDN contents influenced by FTCs in the revised MS of "3. Results [L175-216]".

[L160] "3.1. Soil biogeochemical changes by freeze-thaw cycles" this section includes biogeochemical data and soil aggregate data, so this subheading is not fully correct.

⇨ According to the reviewer's comment, we revised it as: "3.1. Soil biogeochemical and structural changes by freeze-thaw cycles [L176]".

[L161-162] "A higher level of soil $CO_2$ flux was observed in FTC than in CON during the incubation period (Fig. 1a). Accordingly, the cumulative $CO_2$ emission in the FTC soil was 3.6 g m$^{-2}$ hr$^{-1}$, six-times higher than that in the CON soil (Fig. 1b)."
- There are very clear and significant difference shown in these figures, very nice! Maybe include the results from the ANOVA here in the text too to back this up.
⇨ To demonstrate the significance of differences between FTC and CON soils, we have included information on the p-values from ANOVA in the main context [L177-195]. Additionally, to provide more detailed statistical findings, we have added the F-values in Tables 1, 2, 3, and 4.
- The differences observed in Figure 1 could be described in more detail. You do not mention that there are higher respiration rates seen in every day of the incubation. Nor do you mention that you measure these over time. This is all interesting information to include because it show that over 7 days it is a consistent and sustained trend. It is also very interesting that you see an immediate reponse, i.e., on day 1. This suggests that even a single day of FTC in the field may result in greater CO2 emissions. These results should be explored and described in more detail.
⇨ According to the reviewer's point, we added the detailed methods and results of soil respiration (RES) during the soil incubation.
⇨ In [L116-120], we detailed RES measurements during the incubation: "Soil respiration (RES), widely accepted as a proxy for overall microbial activity (Kim and Yoo, 2021; Maikhuri and Rao, 2012; Davidson et al., 1998; Kuzyakov and Domanski, 2000; Lipson and Schmidt, 2004; Raich and Schlesinger, 1992), was estimated by daily measurement of the $CO_2$ flux from soil incubation during the entire incubation period. We collected gas samples from the headspace through a septum using 10 mL syringes (BD Luer-Lok tip, BD Company, Franklin Lakes, NJ, USA) before and after sealing the incubation bottle for 60 mins."
⇨ In [L183-185], we added the explanation on the results of RES analysis: "The mean RES ($RES_{mean}$) in the FTC soil was 42.54 mg m$^{-2}$ hr$^{-1}$, twelve-times higher than that in the CON soil (3.65 mg m$^{-2}$ hr$^{-1}$, $p=0.004$), as shown in Table 2. This is because the RES in FTC soil was significantly higher than in CON soil from the early stages of FTCs ($p<0.05$) and remained consistently higher until the end of the incubation (Fig. S2)."

[L162-164] "On the other hand, no significant differences ($p>0.05$) in microbial extracellular enzyme activities…" include more information about the test performed here, type of test, degrees of freedom, F statistic
⇨ We have shown the F-values related to the results of microbial extracellular enzyme activities in the text and Table 2.

[L165-166] "The DOC and TDN contents in the FTC soil were higher by 29% than those in the CON soil ($p>0.05$).

- "…in the FTC soil were higher by 29% than those in the CON soil…" this does not match with the information provided in table 3. In table 3 DOC and TDN are higher in the controls than in the FTC. DOC and TDN in CON were 659.91 and 39.01, respectively, and only 467.04 and 25.44 in FTC
  ⇨ We clarified the numbers of the Table 1, which were utilized for calculating the difference between FTC and CON soils. In [L177-178], revised as: "The FTC soil exhibited lower DOC and TDN contents by 29% and 35%, respectively, compared to the CON soil ($p<0.001$)."
- "p>0.05" more information on test results
  ⇨ We have shown the detailed statistical findings ($p$- and $F$-value) in Table 1.

[L166-167] "As a proxy for DOC quality, $SUVA_{254}$ was higher but $A_{254}/A_{365}$ was lower in the FTC soil than in the CON soil ($p<0.05$)." What does this mean then, can you describe in more detail what a higher and lower SUVA and A254/A365 indicates?
  ⇨ We have added further explanations regarding the implications of changes in $SUVA_{254}$ and $A_{365}/A_{254}$ in [L179-181]: "The increase in $SUVA_{254}$ indicates an increase in the aromaticity of DOC (Lim et al., 2021), while the decrease in $A_{254}/A_{365}$ reflects an increase in the molecular weight of DOC (Berggren et al., 2018)."

[L167-168] "In contrast, no significant changes ($p>0.05$) in inorganic N ($NH_4^+$-N and $NO_3^-$-N) content were determined because of FTCs." same as above, more information on the test results
  ⇨ We have shown the detailed statistical findings ($p$- and $F$-value) in Table 1.

[L169] "FTCs caused a significant difference in the water-stable aggregate distribution between the FTC and CON soils (Table 4)." but only in 2 of the 4 aggregate/fraction sizes? Granted they make up the greatest proportion of the soil (between 30 - 38% each), but this should be more clearly stated
  ⇨ In the process of reconfirming the statistical analysis results, we found significance at the $p<0.10$ level only in the mass proportion of micro-aggregate. However, the mineral-associated fractions also changed, although not significantly, corresponding to the increased level of micro-aggregates by FTCs. As reviewer's point, we have revised the sentences to more clearly describe that specific-sized aggregate distribution was influenced by FTCs. In [L188-192], revised as: "FTC resulted in minor differences in the mass proportion of micro-aggregates (53–250 μm) and mineral-associated fractions (<53 μm), which account for an average of 35% and 34% of the total in each soil (Table 3). The mass proportion of micro-aggregates marginally increased by 17% in the FTC soil compared to that in the CON soil ($p=0.066$). Although the mineral-associated fractions were insignificantly reduced by FTCs ($p=0.257$), the reduction level (18%) corresponded to the increased distribution of micro-aggregate by FTCs."

[L171-172] "The soil water retention curves showed that the PSD differed significantly between the FTC and CON soils (Table 5)."
- I suggest you show these curves rather than the data in a table
  ⇨ We have added the soil water retention curves in Fig. S3.
- but only in one size fraction? overall it did not have an effect on 3/4 of the fractions. This should be made clearer
  ⇨ The sentences relevant to the PSD results [L192-195] were revised as: "Moreover, FTCs caused a significant difference in the PSD, particularly in the small-sized mesopores (0.2-10 μm), which accounted for 44-45% of the total soil pores (Table 4), as estimated using water retention curves (Fig. S3). Despite the small magnitude of difference, the proportion of small-sized mesopores in the FTC soil exhibited a statistically significant increase compared to that in the CON soil ($p=0.024$)."

[L177-178] "The first two principal components (PCs) accounted for 93.4% of the data variability and could distinctly discriminate the FTC soil from the CON soil (Fig. 2)."
- can you provide more information regarding the PCA fit?
  ⇨ We have added more information for PCA results in [L198-202]: "The first two principal components (PCs) accounted for 93.9% of the total variance, with PC1 clearly clustering the FTC and CON treatments (Fig. 2). PC1 exhibited positive correlations with $SUVA_{254}$, $RES_{mean}$, micro-aggregate, and small-sized mesopores, while it showed negative correlations with DOC, TDN, and $A_{254}/A_{365}$. In Fig. 2, the micro-aggregate was nearly perpendicular to the DOC and TDN contents, $SUVA_{254}$, and $A_{254}/A_{365}$, indicating a weak or no correlation between them."
- Figure 2 shows TOC rather than DOC, and the microaggregate range is 250 - 1000 rather than 53 - 250 um. Also, what do the ellipses represent in figure 2?
  ⇨ We have checked the typos and added the explanation on the ellipses represents in Fig. 2.

[Figure]

**Figure 1: Principal component analysis (PCA) for the FTC and CON soils (n=3). The analysis was performed on the observed variables that showed significant differences between treatments. The input variables were standardized to avoid bias due to their scale. Each arrow to the direction of increase for a given variable and its length indicate the strength of the correlation between the variable and ordination scores. Ellipses show confidence intervals of 95% for each treatment.**

[L181-183] "Lastly, SEM was developed to understand how relationships between soil structural properties and microbial activity contribute to changes in DOC quantity and quality (Fig. 4)." Does the SEM include the FTC and CON data or just the FTC? Also, can you provide the a priori conceptual model of your descriptors used to build this SEM?

⇨ As we answered to the general comments at the beginning of this reply, we decided to perform the MLR analyses instead of SEM to investigate the correlations between soil structural and biogeochemical variables affected by FTCs. Thus, we described how to conduct the MLR analyses in detail. We created plausible relationships based on the results of ANOVA, PCA, and Pearson's correlation analysis; and then refined these relationships with best-fitting regression sets selected from MLR analyses.

⇨ In [L169-174], we have added the methods of MLR analyses: "Finally, multiple linear regression (MLR) analyses were employed to illustrate the mechanisms by which the FTC-influenced soil biogeochemical variables contributed to changes in DOC characteristics. Based on the results of ANOVA, PCA, and Pearson's correlation analysis, we took with all plausible interactions among soil physical and biogeochemical variables significantly affected by FTCs and refined these interactions by finding the best-fitting regression sets from MLR. The MLR analyses were generated using SigmaPlot 13.0."

⇨ In [L206-215], we have added the results of MLR analyses: "Lastly, a conceptual diagram (Fig. 4) was created using MLR analyses (Table 5) to depict the relationships between soil structural properties, microbial activity, and DOC quantity and quality as influenced by FTCs. In Table 5, $RES_{mean}$ and small-sized mesopores were the best-fitting variables for explaining the contents of DOC (Adjusted $R^2$ = 0.911, $p=0.012$) and TDN (Adjusted $R^2$ = 0.869, $p=0.022$). The variance inflation factors (VIFs) resulting from the MLR analyses were <10, indicating no collinearity between $RES_{mean}$ and small-sized mesopores as independent variables. The variables for best representing $SUVA_{254}$ and $A_{254}/A_{365}$ were $RES_{mean}$ (Adjusted $R^2$ = 0.703, $p=0.023$) and small-sized mesopores (Adjusted $R^2$ = 0.878, $p=0.004$), respectively. The addition of micro-aggregate reduced the Adjusted $R^2$ of the best-fitting regression for explaining DOC, TDN, SUVA, and $A_{254}/A_{365}$. Micro-aggregate correlated with $RES_{mean}$ (Adjusted $R^2$ = 0.557, $p=0.054$) and small-sized mesopores (Adjusted $R^2$ = 0.618, $p=0.039$). As a result, we speculated that micro-aggregate indirectly, rather than directly, affected the quantitative and qualitative DOC variables through its correlation with $RES_{mean}$ and SSM, as illustrated in Fig. 4."

⇨ The results of MLR analyses are summarized in Table 5, visualized into Fig. 4.

**Table 1: Multiple linear regression (MLR) analyses the FTC and CON soils. The analysis was performed on the observed variables that showed significant differences between treatments. The independent variables were standardized to avoid bias due to their scale.**

| Dependent variable | # | $R^2$ | $R^2_{adjust}$ | p-value | Predictor | Standardized β coefficient | p-value | VIF |
|---|---|---|---|---|---|---|---|---|
| DOC | 1 | 0.885 | 0.856 | 0.005** | Constant | $-0.664\times10^{-15}$ | - | - |
| | | | | | $RES_{mean}$ | -0.941 | 0.005** | - |
| | 2 | 0.946 | 0.911 | 0.012** | Constant | $-1.116\times10^{-15}$ | - | - |
| | | | | | $RES_{mean}$ | -0.639 | 0.056* | 2.5 |
| | | | | | Small-sized mesopore | -0.390 | 0.162 | 2.5 |
| | 3 | 0.951 | 0.878 | 0.072* | Constant | $-1.513\times10^{-15}$ | - | - |
| | | | | | $RES_{mean}$ | -0.695 | 0.129 | 3.1 |
| | | | | | Small-sized mesopore | -0.464 | 0.260 | 3.4 |
| | | | | | Micro-aggregate | 0.140 | 0.702 | 4.1 |
| TDN | 1 | 0.896 | 0.870 | 0.004** | Constant | $0.017\times10^{-15}$ | - | - |
| | | | | | $RES_{mean}$ | -0.947 | 0.004** | - |
| | 2 | 0.922 | 0.869 | 0.022** | Constant | -0.274 | - | - |
| | | | | | $RES_{mean}$ | -0.752 | 0.060* | 2.5 |
| | | | | | Small-sized mesopore | -0.251 | 0.397 | 2.5 |
| | 3 | 0.928 | 0.821 | 0.011** | Constant | $-0.746\times10^{-15}$ | - | - |
| | | | | | $RES_{mean}$ | -0.818 | 0.135 | 3.1 |
| | | | | | Small-sized mesopore | -0.339 | 0.446 | 3.6 |
| | | | | | Micro-aggregate | 0.167 | 0.707 | 4.1 |
| $SUVA_{254}$ | 1 | 0.763 | 0.703 | 0.023** | Constant | $1.359\times10^{-15}$ | - | - |
| | | | | | $RES_{mean}$ | 0.873 | 0.023** | - |
| | 2 | 0.809 | 0.681 | 0.084* | Constant | $1.751\times10^{-15}$ | - | - |
| | | | | | $RES_{mean}$ | 0.612 | 0.222 | 2.5 |
| | | | | | Small-sized mesopore | 0.338 | 0.458 | 2.5 |
| $A_{254}/A_{365}$ | 1 | 0.902 | 0.878 | 0.004** | Constant | $-2.072\times10^{-15}$ | - | - |
| | | | | | Small-sized mesopore | -0.950 | 0.004** | - |
| | 2 | 0.920 | 0.866 | 0.023* | Constant | $-1.941\times10^{-15}$ | - | - |
| | | | | | Small-sized mesopore | -0.790 | 0.055* | 2.5 |
| | | | | | $RES_{mean}$ | -0.207 | 0.481 | 2.5 |
| RES | 1 | 0.646 | 0.557 | 0.054* | Constant | -2.019 | - | - |
| | | | | | Micro-aggregate | 0.804 | 0.054 | - |
| | 2 | 0.681 | 0.469 | 0.180 | Constant | -0.975 | - | - |
| | | | | | Micro-aggregate | 0.521 | 0.442 | 3.3 |
| | | | | | Small-sized mesopore | 0.340 | 0.605 | 3.3 |
| Small-sized mesopore | 1 | 0.694 | 0.618 | 0.039** | Constant | -3.077 | - | - |
| | | | | | Micro-aggregate | 0.833 | 0.039 | - |

Note: $R^2$, coefficient of determination; $R^2_{adjust}$, Adjusted $R^2$; VIF, variance inflation factor; RES, soil respiration. The asterisks ** and * indicate significant differences between treatments at a $p<0.05$ and $p<0.10$ levels, respectively.

[Figure]

**Figure 2: Conceptual diagram illustrating the response of surface organic soils in the Arctic tundra to the freeze-thaw cycles (FTCs). The blue upward and red downward arrows represent the increase and decrease in the observed variables by FTCs, respectively. The magnitude of correlation and significance was determined by multiple linear regression (MLR) analyses (Table 5).**

**Discussion**

[L190-192] "The seven-successive FTCs increased CO2 emission from the soil during incubation (Fig. 1) and also reduced DOC and TDN contents relative to those observed in the non-treated soil (Table 3). As expected, DOC content was significantly affected by FTCs; however, no decrease but an increase in soil respiration was observed."

- not according to the results section L166
  - ⇨ We apologize for any confusion regarding the results. In [L177-178], we have revised the results with the corrected information: "The FTC soil exhibited lower DOC and TDN contents by 29% and 35%, respectively, compared to the CON soil ($p<0.001$)." Therefore, we have confirmed the relevant parts of discussion were accurate: "The seven-successive FTCs reduced soil DOC and TDN contents compared to the non-treated condition [L219]."
- this is the same as the line above
  - ⇨ We revised the sentences to clarify our intention: "The seven-successive FTCs reduced soil DOC and TDN contents compared to the non-treated condition, aligning with the expectation that the quantitative characteristics of DOC were significantly affected by FTCs (Table 1). These results indicate that FTCs can accelerate the microbial decomposition of labile organic matter (Grogan et al., 2004; Han et al., 2018; Foster et al., 2016; Gao et al., 2021). A proxy for overall microbial activity, RES, remained high throughout the incubation under the influence of FTCs (Fig. S2; Table 2) [L219-223]."

[L200-201] "As the C-rich permafrost thaws, soil C availability in the Arctic tundra would increase dramatically, which can lead to a high risk of $CO_2$ release to the atmosphere (Estop-Aragonés et al., 2020)." Do the authors think then that the increased availability of DOC with increased FTCs is driven by leaching of previously frozen material, or by the breakdown of organic matter in the active layer due to freeze thaw? If the latter, then is the organic matter liable to breakdown most likely of plant origin or microbial necromass? If tundra microbes are resistant to cell lysis at low temperatures, is the necromass of them also resistant? In this study, do the results reflect more the organic surface layer or the C being released by permafrost thaw and active layer thickening. My interpretation is that the material that is labile in this study is not previous frozen material, but rather organic matter accumulated at the surface that consists of plant material and non living microbial biomass that was never part of the permafrost lens. I would suggest to the authors to explore this approach in discussing their results, and compare the effect of FTCs in tundra sites to non tundra sites. In non tundra sites, do FTCs similarly effect the organic soil layer as shown in this study? This to me is very interesting as it shows how surface warming will impact carbon dynamics in the active layer. In order to assess how carbon released due to thawing is impacted by FTCs a different study design would be required.

  - ⇨ We appreciate the reviewer's sharp critics on our rough writing. As the reviewer pointed out we focused on changes caused by FTCs on the active layer composed of organic materials. Thus, we revised the sentences in [L226-236]: "Soil microbes in the Arctic tundra could survive at temperatures below -7 to -11 ℃ (Lipson et al., 2000; Männistö et al., 2009; Lipson and Monson, 1998), the general threshold for microbial cell lysis in non-tundra environments (Gao et al., 2018a, 2021; Song et al., 2017). The microbes that can survive under these freezing conditions actively play a role in decomposing available DOC in the surface organic layer during thaw phases in FTCs. In addition, the top organic layer was composed of a higher quality plant-derived organic matter compared to the underlying mineral layer in Council, Alaska, a similar ecosystem as our study site (White et al., 2004). Thus, the biologically labile DOC could be available in the surface organic layer (Gao et al., 2018b). Hence, decreases in DOC associated with activated microbial activities following FTCs suggest that responses of the DOC in the organic layer to FTCs would be crucial in affecting the tundra C cycle under Arctic warming. More frequent FTCs and a longer thawing length in tundra soils with warming could enhance soil C availability in the active layer of the Arctic terrestrial ecosystems, leading to a high risk of $CO_2$ being released into the atmosphere (Estop-Aragonés et al., 2020)."

⇨ As the reviewer mentioned, organic matter accumulated in the active layer of the Arctic tundra mainly originates from above- and below-ground residues of vegetation, such as cellulose, lignin, and proteins. In particular, White et al. (2004) reported that the surface organic layer of the Alaskan acidic tundra, the same region of this study, was composed of higher quality plant-derived organic matter compared to the mineral layer. Therefore, we believe that organic matter in the surface organic layer can produce free-particulate or extractable/leachable forms through the FTCs, eventually leading to the decomposition of biologically labile DOC in the topsoil layer (Gao et al., 2018).

[L202-206] "Meanwhile, FTCs did not significantly change the activities of extracellular enzymes (Table 2), which are released by soil microbes to obtain C and N from recalcitrant soil organic matter such as cellulose, chitin, polypeptides, and lignin (Sinsabaugh, 2010; Liao et al., 2022). This may be because soil microbes preferentially utilize simple compounds that do not require 205 enzymes for degradation in the process of DOC decomposition enhanced by FTCs (Foster et al., 2016; Gao et al., 2021; Perez-Mon et al., 2020)."

- The enzyme assays used only measure potential enzyme activities, not actual rates of enzymatic degradation. While max potential rates may not differ, actual rates may do so.

  ⇨ We agreed with the reviewer's comments. Thus, we added the plausible explanation for the non-significant changes, that is, measuring potential activities might not reflect the actual rates: "However, the enzyme activities in this study were measured under laboratory conditions with sufficient substrate supplies and suitable environment; therefore, these potential activities may not properly reflect actual microbial enzyme activities under these study conditions. Despite this inherent limitation, we argue there is a non-significant in measured enzyme activity caused by FTCs, as soil microbes preferentially utilize simple compounds that do not require enzymes for degradation in DOC decomposition enhanced by FTCs (Foster et al., 2016; Gao et al., 2021; Perez-Mon et al., 2020). These results were evidenced by the different DOC quality between the FTC and CON soils (Table 1). The DOC quality indices, $SUVA_{254}$ and $A_{254}/A_{365}$, significantly differed between the FTC and CON soils, indicating that complex substrates with high aromaticity and molecular weight remained in the dissolved organic matter after successive FTCs (Berggren et al., 2018; Yang et al., 2019) [L239-447]."

- Is 7 days a long enough time to identify a difference in the production of extracellular enzymes?

  ⇨ According to the reviewer's point, no changes in extracellular enzyme activities can be a short duration of experiments. However, Gao et al. (2021, Soil Biology and Biochemistry), who performed a meta-analysis of FTC studies, reported enzyme activities influenced by FTCs could be observed within a few days or hours. Additionally, the immediate response of RES (Fig. S2) suggests that the lack of responses in enzymatic activities cannot be solely attributed to the duration of the experiment.

- Is there evidence that the vegetation present at the site produces a wide variety of these simple compounds? My thought would be that tundra vegetation organic matter that is high in lignin and other complex compounds that are difficult to break down.

  ⇨ As the reviewer mentioned, organic matter accumulated in the active layer of the Arctic tundra mainly originates from above- and below-ground residues of vegetation, such as cellulose, lignin, and proteins. In particular, White et al. (2004) reported that surface organic layer of the Alaskan acidic tundra, the same region of this study, has higher quality plant-derived organic matter compared to the mineral layer. This can allow for the existence of biologically labile DOC in the topsoil layer (Gao et al., 2018). In this study, our results showed that the amount of organic-layer DOC content was decreased by FTCs, but its aromaticity and molecular weight were increased. This result indicates that simple organic compounds with low aromaticity and low molecular weight were used relatively preferentially in the DOC decomposition process. Thus, our data on DOC quality indices ($SUVA_{254}$ and $A_{365}/A_{254}$) showed DOC in the FTCs-treated soil had higher aromaticity and molecular weight compared to that in the non-treated soil. Thus, we still argue that non-significant changes in extracellular enzyme activities could be partially caused by the preferential use of simple compounds available in the organic active layer.

[L215] "FTCs increased the mass proportion of macro-aggregates compared with those observed in the non-treated soil (Table 4)." This is not supported by the data in table 4.

  ⇨ We apologize for any confusion regarding the results due to the typos. In [L255-256], revised as: "FTCs caused an increase in micro-aggregate (53-250 μm) and a corresponding decrease in mineral-associated fractions (<53 μm), despite low significance levels (Table 3)."

[L216-217] "This might be related to the lower water content in the FTC soil (0.63±0.03 g water/g soil) than in the CON soil (0.71±0.01 g water/g soil)…" is this data shown elsewhere?

  ⇨ We have deleted the relevant sentences. This is because our experimental design cannot show the effect FTCs on soil body or aggregate breakdown under different soil moisture conditions. Therefore, we focused more on the explanation of micro-aggregate enhanced by FTCs.

[L228-229] "Soil micro-aggregates enhanced by FTCs affected DOC quantity and quality mainly through changes in the microbemediated mechanism rather than the physical mechanism (Fig. 4)." I do not think your SEM shows this, or the data can show this. CO2 is a result of DOC mineralization not a driving factor in DOC concentrations. DOC concentrations may change after mineralization

  ⇨ As the reviewer states, $CO_2$ is a degradation product of DOC. However, soil $CO_2$ flux has been widely accepted as a proxy of overall activity of microorganisms degrading SOM. We elaborated on the RES concept by describing it in "2. Materials and methods" and adding relevant references. In [L116-118], "Soil respiration (RES), widely accepted as a proxy for overall microbial activity (Kim and Yoo, 2021; Maikhuri and Rao, 2012; Davidson et al., 1998; Kuzyakov and Domanski, 2000; Lipson and Schmidt, 2004; Raich and Schlesinger, 1992), was estimated by daily measurement of the $CO_2$ flux from soil incubation during the entire incubation period."

[L233-234] "Consequently, our findings suggest that the improvement of soil structure and function by FTCs contribute to DOC decomposition by soil microbes, thereby reducing the DOC content and increasing the DOC aromaticity in the FTC soil (Fig. 4)." on the micro aggregate scale only

⇨ Admitting the reviewer's point, we specifically mentioned the scale of soil structure: "Consequently, our findings suggest that soil structural improvement, at the micro-aggregate scale, by FTCs contributes to DOC decomposition by soil microbes, thereby resulting in reduced DOC content and increased DOC aromaticity in the FTC soil (Fig. 4) [L269-271]."

[L235-237] "On the other hand, FTCs can affect DOC quantity through physical mechanisms as well, such as soil aggregation and pore formation. We found that the significant difference in soil PSD affected by FTCs was in the volume of the small-sized mesopores (Table 5)." I think this is a much more likely explanation of the processes driving the changes in DOC concentrations

⇨ In the MLR analyses, we found that both RES and small-sized mesopores significantly and directly contributed to the quantity and quality of DOC; however, micro-aggregates did not show any direct impacts on DOC characteristics. In particular, the proxy of microbial activity, RES, contributed more to explaining DOC properties than small-sized mesopores. Therefore, we preferentially described and interpreted a pathway in which FTCs-affected soil aggregates alter DOC properties by enhancing microbial activity.

**Conclusions**

[L253-255] "We found that the following seven variables differed significantly by FTCs: soil respiration, DOC and TDN contents, two DOC quality indices, micro-aggregate distribution, and small-sized mesopores volume." name the indices

⇨ In [L290-292], revised as: "We found that the following seven variables differed significantly after FTCs: soil respiration (RES), DOC and TDN contents, two DOC quality indices ($SUVA_{254}$ and $A_{365}/A_{254}$), micro-aggregate (53–250 μm) distribution, and small-sized mesopore (0.2–10 μm) proportion."

Figure 4
- I am not sure that this casual pathway is sound. How does cumulative CO2 emissions impact DOC? CO2 is produced from the mineralization of DOC, so this pathway moves in the opposite direction. True, DOC will be lower in soils with high CO2, but the CO2 emissions are not driving DOC concentrations. Cumulative CO2 emissions should be the final outcome in the SEM, not a casual pathway. This pathway should be removed

⇨ Consistent with responses to previous comments, we considered $CO_2$ flux as a proxy of overall activity of microorganisms degrading SOM. We elaborated on the RES concept by describing it in "2. Materials and methods" and adding relevant references. In [L116-118], "Soil respiration (RES), widely accepted as a proxy for overall microbial activity (Kim and Yoo, 2021; Maikhuri and Rao, 2012; Davidson et al., 1998; Kuzyakov and Domanski, 2000; Lipson and Schmidt, 2004; Raich and Schlesinger, 1992), was estimated by daily measurement of the $CO_2$ flux from soil incubation during the entire incubation period."

- The prior model that this SEM is based off should also be included with each hypothesis behind each pathway

⇨ As we answered to the general comments at the beginning of this reply, we decided to perform the MLR analyses instead of SEM to investigate the correlations between soil structural and biogeochemical variables affected by FTCs. Thus, we described how to conduct the MLR analyses in detail. We created plausible relationships 
[revised manuscript text omitted]

[Figure]

**Figure S3. Water release curves and pore size distribution in the FTC and CON soils.**